# Induction of Sertoli-like cells from human fibroblasts by NR5A1 and GATA4

Jianlin Liang[1,2], Nan Wang[1], Jing He[1], Jian Du[1], Yahui Guo[1], Lin Li[1], Wenbo Wu[3], Chencheng Yao[4], Zheng Li[4], Kehkooi Kee[1,2,5]*

[1]Center for Stem Cell Biology and Regenerative Medicine, Department of Basic Medical Sciences, School of Medicine, Tsinghua University, Beijing, China; [2]Tsinghua-Peking Center for Life Sciences, School of Life Sciences, Tsinghua University, Beijing, China; [3]National Institute of Biological Sciences, Beijing, China; [4]Department of Andrology, the Center for Men's Health, Urologic Medical Center, Shanghai Key Laboratory of Reproductive Medicine, Shanghai Jiao Tong University School of Medicine, Shanghai General Hospital, Shanghai, China; [5]Beijing Advanced Innovation Center for Structural Biology, School of Life Sciences, Tsinghua University, Beijing, China

**Abstract** Sertoli cells are essential nurse cells in the testis that regulate the process of spermatogenesis and establish the immune-privileged environment of the blood-testis-barrier (BTB). Here, we report the in vitro reprogramming of fibroblasts to human induced Sertoli-like cells (hiSCs). Initially, five transcriptional factors and a gene reporter carrying the *AMH* promoter were utilized to obtain the hiSCs. We further reduce the number of reprogramming factors to two, NR5A1 and GATA4, and show that these hiSCs have transcriptome profiles and cellular properties that are similar to those of primary human Sertoli cells. Moreover, hiSCs can sustain the viability of spermatogonia cells harvested from mouse seminiferous tubules. hiSCs suppress the proliferation of human T lymphocytes and protect xenotransplanted human cells in mice with normal immune systems. hiSCs also allow us to determine a gene associated with Sertoli cell only syndrome (SCO), CX43, is indeed important in regulating the maturation of Sertoli cells.

*For correspondence:
kkee@tsinghua.edu.cn

Competing interests: The authors declare that no competing interests exist.

## Introduction

Sertoli cells are the first somatic cell type to differentiate in the testis and the only somatic cell type inside the seminiferous tubules. Sertoli cells play a critical role in directing testis morphogenesis and the creation of an immune-privileged microenvironment, which is required for male germ cell development. During early gonad development, male somatic cells express the male sex-determining gene *SRY*, which directs the sex-specific vascular development and seminiferous cord formation (*Bott et al., 2006*; *Brennan et al., 2003*; *Koopman et al., 1990*) via the initiation of a cascade of genes, including *SOX9, FGF9, AMH* and *PGD2* (*Barrionuevo et al., 2009*; *Moniot et al., 2009*). *NR5A1* (or *SF1*), *GATA4*, and *WT1* are major transcriptional factors that direct somatic cells to become fetal Sertoli cells (*Rotgers et al., 2018*). Five transcriptional factors have been demonstrated to successfully reprogram mouse fibroblasts to Sertoli cells (*Buganim et al., 2012*). The expanding fetal Sertoli cells and another type of testicular somatic cell (i.e., peritubular cells) regulate the final organization and morphogenesis of the developing gonad into a testis (*Griswold, 1998*; *McLaren, 2000*).

Sertoli cells are the pivotal somatic cell regulators inside the seminiferous cord. Sertoli cells embed male germ cells during all differentiating stages and provide immunological, nutritional and structural support for germ cell development (*Oatley and Brinster, 2012*). Sertoli cells secrete the

growth factors and cytokines needed for proper spermatogenesis, including the maintenance of spermatogonial stem cells, meiosis initiation of spermatocytes, and maturation of spermatozoa (*Hai et al., 2014*). Furthermore, Sertoli cells have the unique ability to modulate immunoreactions that protect the developing germ cells from immunological attacks. The immune-privileged potential of Sertoli cells has been utilized in many allo- and xeno-grafts to reduce the immune response in the field of cell transplantation (*Kaur et al., 2015*; *Mital et al., 2010*; *Valdés-González et al., 2005*). Preclinical studies have transplanted Sertoli cells with various other cell types for the treatment of diabetes, neurodegenerative diseases, Duchenne muscular dystrophy, skin allografts and other diseases (*Luca et al., 2018*).

Recently, co-cultures of differentiated rodent primordial germ cells and neonatal testicular somatic cells have successfully enabled meiosis completion and round spermatid formation in vitro (*Zhou et al., 2016*), highlighting the potential use of testicular somatic cells in the field of reproductive medicine although more experimental validations and improvements are needed. Human pluripotent stem cells have been differentiated to spermatid-like cells (*Easley et al., 2012*; *Kee et al., 2009*), but the co-culturing of stem cells with Sertoli cells could enhance the efficiencies of obtaining functional male gametes. However, the procurement of human Sertoli cells is not feasible because of biological and ethical constraints. The availability of donated Sertoli cells is limited, and expanding the limited number of human Sertoli cells in vitro remains a challenge (*Chaudhary et al., 2005*; *Kulibin and Malolina, 2016*). Therefore, the generation of Sertoli cells from fibroblasts could alleviate these issues and fulfill the basic research and clinical demands.

Direct lineage reprogramming has been considered a promising strategy for obtaining functional cell types with lower teratoma risks than directed differentiation of pluripotent stem cells (*Cherry and Daley, 2012*; *Xu et al., 2015*). The induction of cell type conversion between divergent lineages has been achieved using combinations of lineage-specific transcription factors (*Hendry et al., 2013*; *Huang et al., 2014*; *Nam et al., 2013*; *Yamanaka and Blau, 2010*). Fibroblasts are common cells in animal connective tissues that can be conveniently obtained from patients. Therefore, fibroblasts are often used as initiating cells in many lineage reprogramming experiments. The direct reprogramming of Sertoli cells from fibroblasts has been demonstrated in mouse (*Buganim et al., 2012*), but the direct lineage conversion of human Sertoli cells from fibroblasts has not been described. Here, we report the efficient induction of human Sertoli cells (hiSCs) from both primary human fibroblasts and fibroblasts derived from human embryonic stem cells (hESCs). These hiSCs exhibit an epithelial morphology, lipid droplet accumulation, and transcriptomes similar to those of primary Sertoli cells; sustain the growth of mouse spermatogonia cells; and perform immune-privileged function during transplantation experiments.

Connexin 43 (CX43) is a predominant gap junction protein expressed in BTBs that affects the maturation of Sertoli cells and spermatogenesis (*Brehm et al., 2007*; *Gerber et al., 2016*; *Sridharan et al., 2007*; *Weider et al., 2011*). The deletion of *CX43* in Sertoli cells, but not germ cells, causes infertility in mice (*Brehm et al., 2007*; *Günther et al., 2013*). The absence of CX43 expression in human Sertoli cells is associated with Sertoli cell-only syndrome (SCO) and impaired spermatogenesis in male patients (*Brehm et al., 2002*; *Defamie et al., 2003*), but whether the deletion of *CX43* directly affects the characteristics of human Sertoli cells has not been demonstrated. Utilizing our in vitro hiSC model, we demonstrate that the deletion of *CX43* affects the transcriptome profile and maturation of hiSCs.

## Results

### Testing the reprogramming capability of five putative transcriptional factors

Based on the reprogramming capability of the transcriptional factors reported in a mouse study (*Buganim et al., 2012*), we first tested the reprogramming capabilities of the human homologs of the five transcriptional factors (5TFs: NR5A1, GATA4, WT1, SOX9 and DMRT1) to convert human fibroblasts to hiSCs. All five human homologs were correctly cloned into lentiviral vectors and expressed at high levels as verified by immunofluorescent staining of HPFs and transcriptional level in human embryonic kidney cells (*Figure 1A* and *Figure 1—figure supplement 1A*). After the lentiviral transduction with all five factors and culturing in selective medium for 5 days (*Figure 1B*), many

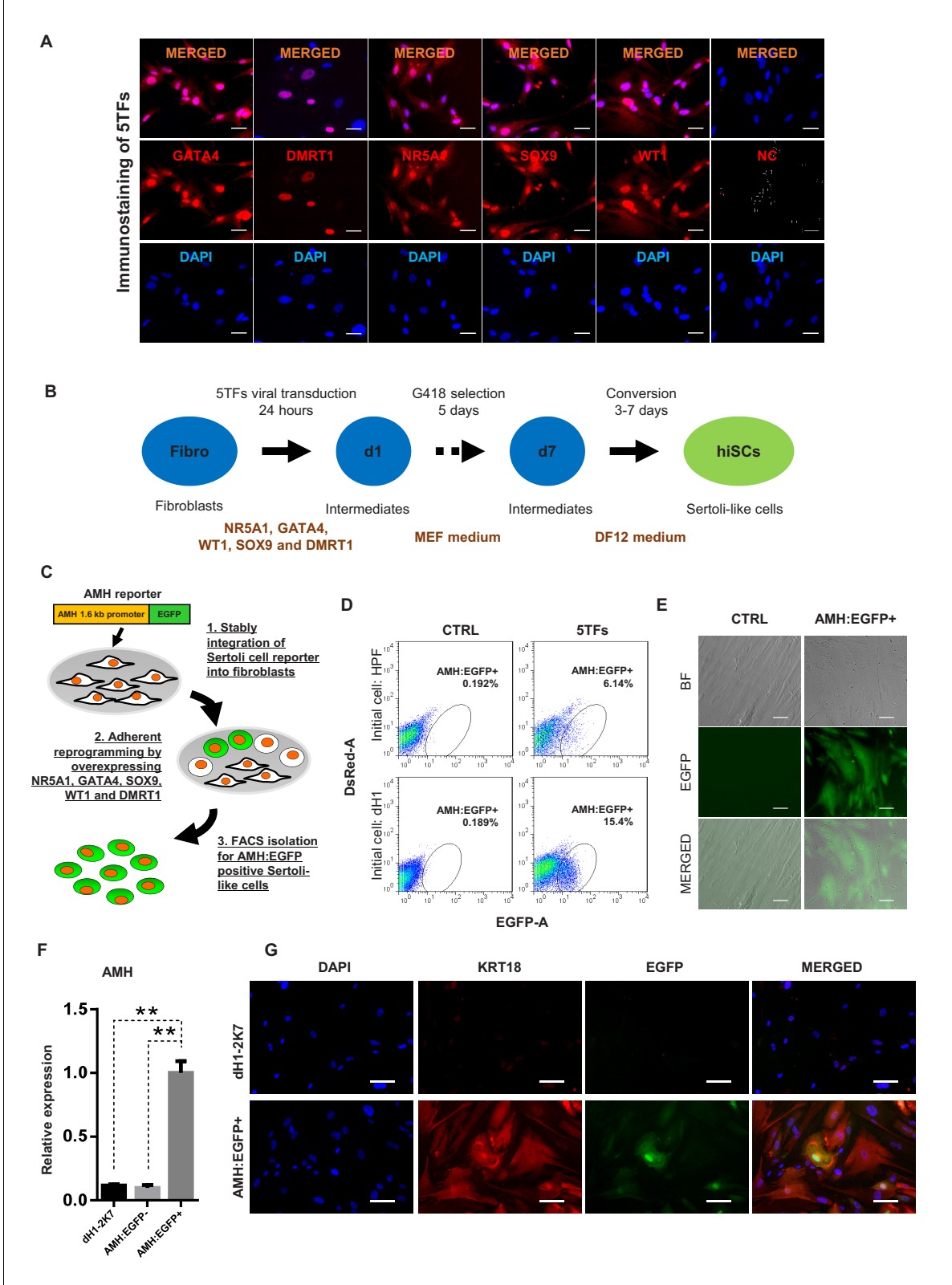

**Figure 1.** Induction of Sertoli-like cells (hiSCs) from human fibroblasts. (**A**) Immunostaining of NR5A1, GATA4, WT1, SOX9 and DMRT1 (red signal) in fibroblasts (HPF) 3 days post infection. DAPI (blue signal) was used to indicate nucleus. Scale bar = 25 μm. NC represents HPF stained with only secondary antibody. (**B**) Experimental design for the reprogramming of human Sertoli-like cells (hiSCs). Human fibroblasts were infected by lentivirus carrying human transcription factors: NR5A1, GATA4, WT1, SOX9 and DMRT1 (5TFs). The cells were cultured in MEF medium for 1 day after infection

*Figure 1 continued on next page*

*Figure 1 continued*

and followed by G418 selection for 5 days, then changed to DMEM/F12 medium. hiSCs were characterized 10–14 days post viral transduction. (**C**) Schematic protocol for the enrichment of Sertoli-like cells (hiSCs). An AMH:EGFP reporter was integrated to the fibroblasts for hiSCs quantification and isolation. (**D**) FACS analysis of AMH:EGFP+ cell at day 10 after 5TFs infection. The percentage of EGFP+ cell was used to determine the reprogramming efficiency. Two types of human fibroblasts (HPF and dH1) were tested. Cells infected by p2k7 empty virus were used as negative control (CTRL). (**D**) Morphology of re-plated AMH:EGFP+ hiSCs after FACS. Scale bar = 20 µm. (**F**) The mRNA level of AMH was enriched in AMH:EGFP + hiSCs, as compared to dH1 infected by p2k7 empty virus (dH1-2K7) and AMH:EGFP-. n = 3, technical replicates of ~5 × 10$^4$ cells were collected by FACS. All data are presented as means ± SD. *p<0.05, **p<0.01, Student's t-test.>3 independent experiments were carried out. (**G**) Immunofluorescent staining of KRT18 in AMH:EGFP+ hiSCs and dH1 infected by p2k7 empty virus (dH1-2K7) after FACS. Scale bar = 100 µm.

The online version of this article includes the following figure supplement(s) for figure 1:

**Figure supplement 1.** Reprogramming effect of 5F on human primary fibroblasts.
**Figure supplement 2.** Reprogramming effect of 5F on differentiated hESC H1.
**Figure supplement 3.** FACS analysis of AMH:EGFP+ population in adult Sertoli cells 10 days after AMH:EGFP reporter virus infection.

HPFs started to transform from the typical elongated morphology of fibroblasts into the squamous morphology that typically appears in epithelial cells (*Figure 1—figure supplement 1B*). The analysis of the transcriptional expression showed that genes enriched in mesenchymal-to-epithelial transition (*Buganim et al., 2012*; *Li et al., 2011*; *Samavarchi-Tehrani et al., 2010*), including *MUC1*, *CLDN1*, *CLDN7*, *CLDN11 and TJP1*, exhibited increased expression in this mixed population of transformed HPFs (*Figure 1—figure supplement 1C*). In addition, the transcriptional expression of genes enriched in Sertoli cells, such as *AR, KRT18, CLU, PTGDS, SCF, BMP4 and INHA*, exhibited increased expression in this mixed population of transformed HPFs (*Figure 1—figure supplement 1D*). To determine whether these 5TFs were able to reprogram other fibroblast sources, we derived human fibroblast-like cells from hESC line H1 (dH1) and reprogrammed these cells as described for the HPFs. The dH1 morphology resembled that of fibroblasts, and no detectable expression of pluripotent markers was observed, but the expression of many markers of fibroblasts was observed (*Figure 1—figure supplement 2A, B and C*). After the transduction of the 5TFs, dH1 underwent a fibroblast to epithelial transformation similar to that observed in the HPFs (*Figure 1—figure supplement 2D*), suggesting that the 5TFs can transform both types of fibroblasts into epithelial-like cells and increase the expression of Sertoli cell markers.

To isolate and enrich the hiSCs in the mixed population of reprogrammed fibroblasts, we constructed a gene reporter system utilizing the 1.6 kb promoter region of *AMH*, which is a gene specifically expressed in Sertoli cells (*Franke et al., 2004*), connected to EGFP (*Figure 1C*). Recent report of single-cell transcriptome analysis also confirms that AMH is specifically expressed in Sertoli cells but not other somatic cells of adult human testis (*Wang et al., 2018*). We constructed a gene reporter carrying *AMH* promoter fusing to EGFP gene and transduced the reporter into adult human Sertoli cells to confirm EGFP expression (*Figure 1—figure supplement 3*). When the lentivector carrying the reporter, AMH:EGFP, was stably integrated into HPFs and dH1, none of the cells would express EGFP without transcriptional induction. After the transduction and selection of the fibroblasts, some reprogrammed cells were expected to express EGFP to allow us to isolate them by fluorescence-activated cell sorting (FACS). We found that both HPF and dH1 reproducibly yielded a clear AMH:EGFP-positive population after 10 days of transduction with the 5TFs (*Figure 1D*). Intriguingly, the AMH:EGFP+ population in the dH1 group (~15%) was much higher than that in the HPF group (~6%), indicating that the conversion of the dH1 cells was more efficient. The AMH:EGFP+ cells were isolated by FACS and adhered to culture dishes and exhibited an epithelial morphology (*Figure 1E*). We verified that the endogenous AMH expression was activated because the expression level of the *AMH* gene was significantly upregulated in the AMH:EGFP+ cell population compared to that in the control dH1 cells (CTRL) and AMH:EGFP- cells (*Figure 1F*). The specific expression of endogenous *AMH* in AMH:EGFP+ cells validated the function of the reporter to isolate the reprogrammed Sertoli cells from the mixed populations which might contain other cell types. Moreover, an epithelial marker expressed in Sertoli cells, that is, KRT18, was used to validate the transformation of the fibroblasts to Sertoli cells as one kind of epithelial cells. The cytoskeleton pattern of KRT18 expression was observed in the AMH:EGFP+ cells but not the control dH1 cells (*Figure 1G*).

## Whole-genome transcriptional profiling of hiSCs resembling adult Sertoli cells

To determine whether hiSCs reprogrammed with the 5TFs (5F-hiSCs) are similar to human Sertoli cells, we compared the transcriptomes of the AMH:EGFP+ 5F-hiSCs, dH1 cells infected with p2k7 empty virus (dH1-2K7) as negative controls, and primary adult Sertoli cells (aSCs) from human biopsy samples (*Wen et al., 2017*). We focused our analysis on the differentially expressed genes (DEGs,>2 fold change, p-value<0.05) between 5F-hiSCs and dH1-2K7 and between aSCs and dH1-2K7. In total, 7533 genes were differentially expressed between 5F-hiSCs and dH1-2K7, including 4528 upregulated genes and 3005 downregulated genes (*Figure 2A and B*). Additionally, 5377 genes were differentially expressed between aSCs and dH1-2K7, including 3343 upregulated genes and 2034 down-regulated genes. The Venn analysis showed that 3626 genes were shared among the DEGs in both hiSCs and aSCs, accounting for approximately 67% of the DEGs in aSCs. Among this shared group of DEGs (CO-DEGs), 1973 genes were upregulated, while 1314 genes were down-regulated in both the hiSCs and aSCs (*Figure 2C*), indicating that the trends in transcriptional expression between the hiSCs and aSCs were the same in these genes. The cluster analysis of dH1-2K7, hiSCs and aSCs also showed that the CO-DEGs had a similar expression pattern between the hiSCs and aSCs, and consistency was observed between duplicate samples (*Figure 2D*). The Gene Ontology (GO) analysis of the CO-DEGs showed that among the 1973 upregulated genes, many genes were involved in the regulation of cell communication, regulation of immune response processes, response to hormones, and lipid metabolic process, whereas among the 1314 down-regulated genes, many genes were involved in the mitotic cell cycle and microtubule-based processes (*Figure 2E*). These changes in gene expression indicated that the hiSCs acquired unique cellular characteristics that were distinct from the original fibroblasts. To further confirm that the AMH:EGFP + 5F-hiSCs have the signature of Sertoli cells, we examined the expression of several Sertoli cell markers, including *CLU, NCAM2, DHH, ERBB4, INHB, INHA, SHBG, GATA6, CDKN1B, TGFα* and *LMMA3*, by quantitative PCR (qPCR). Compared to the control cells, all Sertoli cell markers were highly enriched in the AMH:EGFP+ 5F-hiSCs (*Figure 2F*). Taken together, the transcriptional profile of the AMH:EGFP+ 5F-hiSCs resembled that of the aSCs, and many Sertoli cell markers were expressed.

## NR5A1 and GATA4 are sufficient to reprogram fibroblasts to hiSCs as 5F-hiSCs

Although the 5TFs yielded the AMH:EGFP+ hiSCs, the combination of all 5TFs may not be necessary to reprogram fibroblasts to hiSCs. Therefore, we used fewer transcription factors to generate hiSCs and compared the percentage of AMH:EGFP+ cells in all 31 combinations of NR5A1, GATA4, SOX9, WT1 and DMRT1. The FACS results indicated that 16 combinations of transcriptional factors yielded varying levels of AMH:EGFP+ cells after 10 days of reprogramming (*Figure 3A*). NR5A1 was the only common factor found in all 16 combinations in AMH:EGFP+ cells. This transcriptional factor alone generated approximately 3.79% of AMH:EGFP+ cells, and the combination of all 5TFs produced the highest percentage of AMH:EGFP+ cells. Surprisingly, the combinations with NR5A1 and GATA4 generated as many AMH:EGFP+ cells as all 5TF combined. Moreover, all combinations containing NR5A1 and GATA4 resulted in a similar level higher than the combinations with NR5A1 (*Figure 3B*). The AMH:EGFP+ cells generated by 2TFs and 5TFs showed similar morphologies, including a large cell body with an epithelial morphology, and expressed KRT18 (*Figures 1G* and *3C*).

Then, we analyzed and compared the transcriptome of the 2F-hiSCs and with the transcriptome of the aSCs and 5F-hiSCs. We identified the common DEGs among the dH1/aSCs, dH1/5F-hiSCs (dH1), dH1/2F-hiSCs (dH1) and dH1/5F-hiSCs (HPF) and performed hierarchical clustering using their FPKM values. The analysis revealed that the transcriptome profiles of the 2F-hiSCs were more similar to the profiles of the 5F-hiSCs and adult Sertoli cells (aSCs) than to the dH1 and HPF profiles (data not shown). To identify the putative signature genes similar among the 2F-hiSCs, 5F-hiSCs and aSCs, we generated a heat map of the 1638 CO-DEGs and carried out gene correlation clustering. Notably, the differentially expressed genes were grouped into three groups of 512, 689 and 437 genes (*Figure 3D*). The Gene Ontology analysis showed that many of the 512 highly expressed genes mostly shared by the 2F-hiSCs, 5F-hiSCs and aSCs were involved in reproductive structure

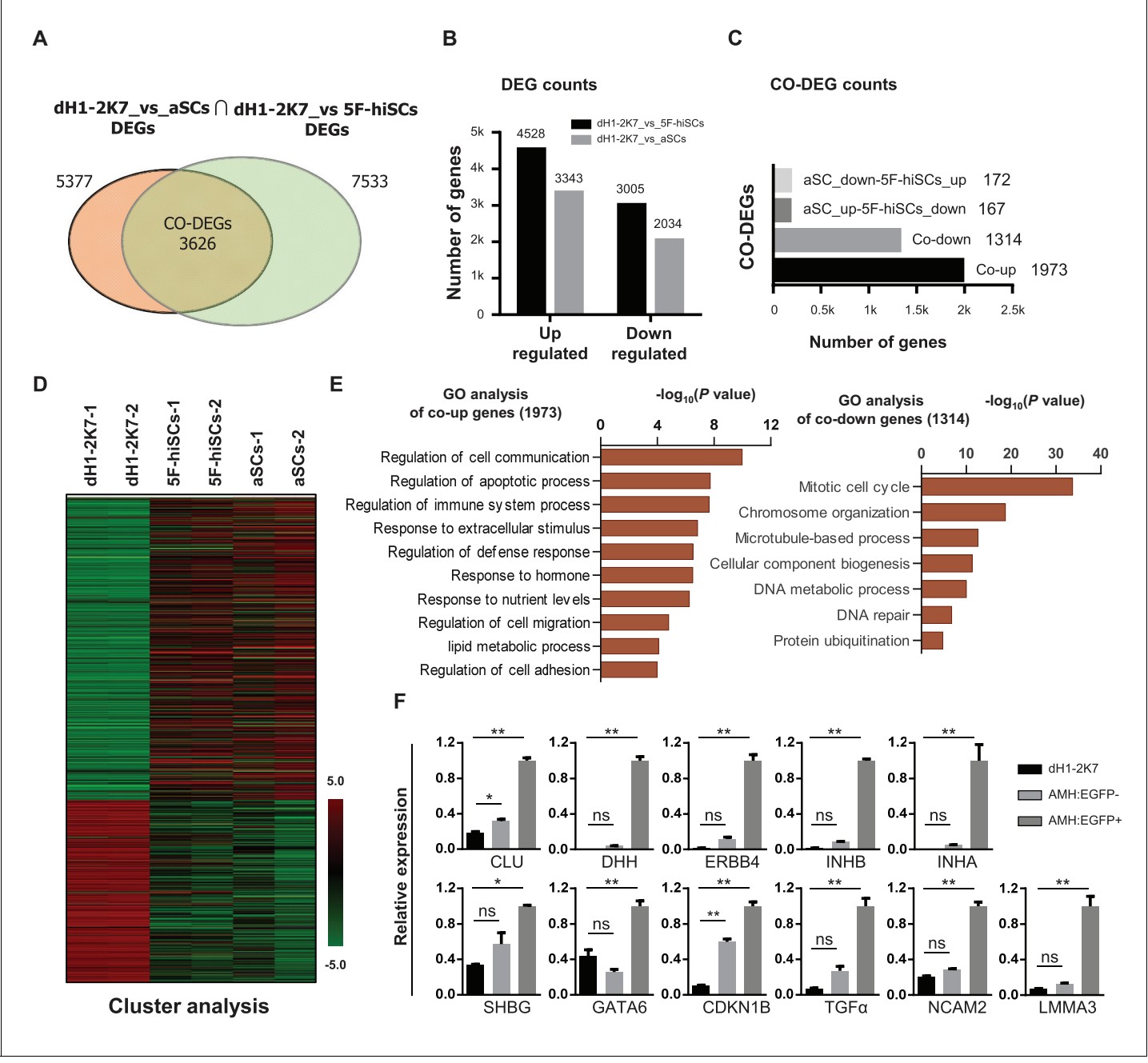

**Figure 2.** Whole genome transcriptional profiling of AMH:EGFP+ hiSCs. (**A**) Venn diagram to display differentially expressed genes (DEGs, FPKM value, fold change >2) between dH1-2K7_vs_aSCs (n = 2) and dH1-2K7_vs_5F-hiSCs (n = 2). Intersection part represents the number of co-differentially expressed genes (CO-DEGs). (**B**) Summary of DEGs in (**A**). X axis represents upregulated DEGs and downregulated DEGs. Y axis represents DEGs numbers. Comparisons of different sample sets were represented by different colors. Black bar represents dH1-2K7_vs_5F-hiSCs. Gray bar represents dH1-2K7_vs_aSCs. (**C**) Summary of CO-DEGs in (**A**). X axis represents CO-DEGs number. Black bar represents co-upregulated DEGs. Gray bar represents co-downregulated DEGs. Dark gray bar and light gray bar represent CO-DEGs in opposite trend, respectively. (**D**) Heat map of gene expression of dH1-2K7, 5F-hiSCs and aSCs (n = 2 in each group). Red indicates upregulated expression, green indicates downregulated expression. (**E**) Functional enrichment analysis, biological processes of 1973 and 1314 differentially expressed genes were showed. (**F**) The mRNA level of Sertoli cell markers in dH1-2K7, AMH:EGFP- cells and AMH:EGFP+ cells, as measured by qPCR. Relative expression was normalized to dH1-2K7. GAPDH was used as the housekeeping gene. All data are presented as means ± SD, n = 2, *p<0.05, **p<0.01, Dunnett's test. three independent experiments were carried out.

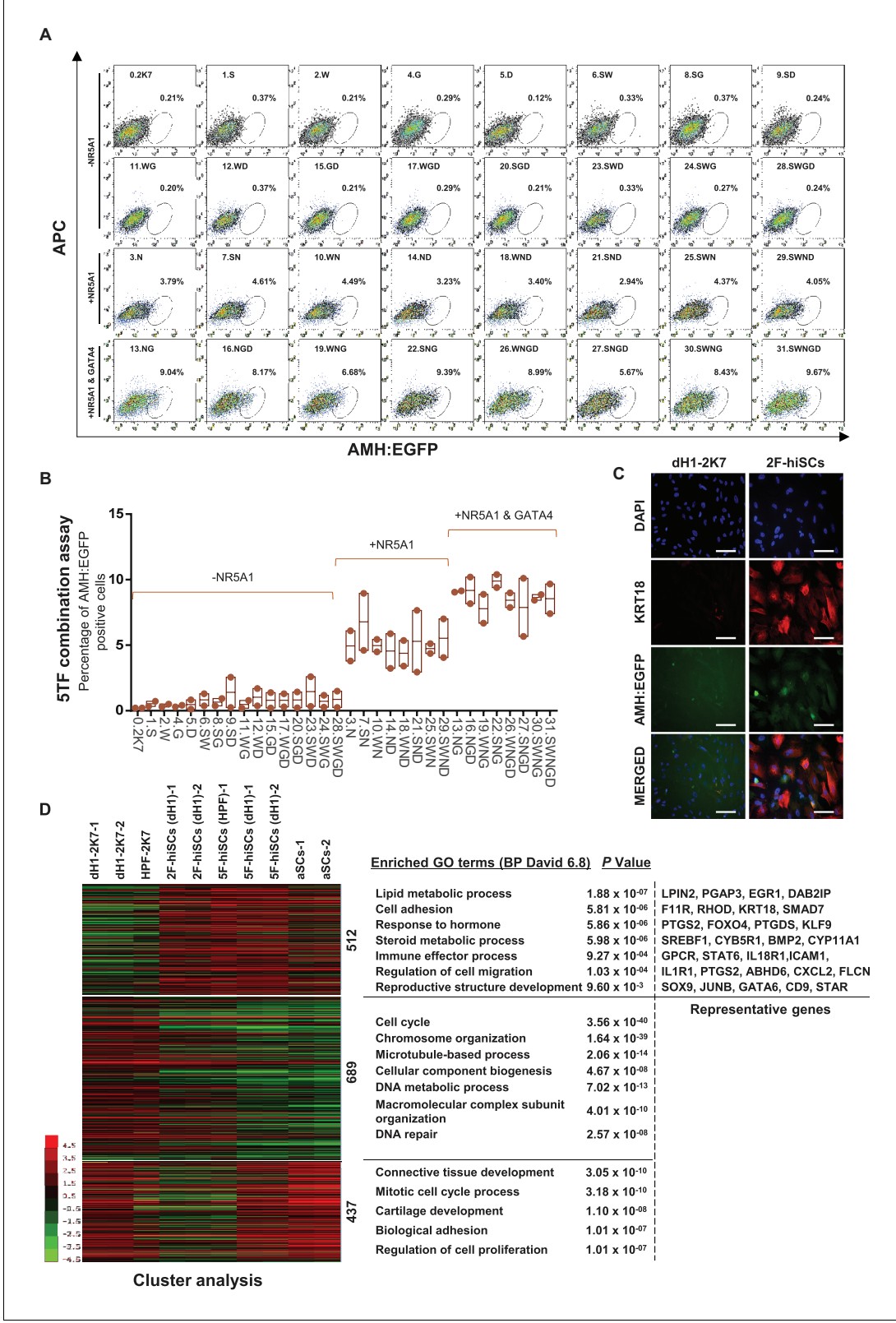

**Figure 3.** NR5A1 and GATA4 are sufficient to derive hiSCs. (**A**) Representative FACS results of different combinations of NR5A1, GATA4, WT1, SOX9 and DMRT1 for the induction of AMH:EGFP+ cells. dH1 fibroblasts were transduced with the indicated factors and reprogrammed for 10 days. The combinations were divided into three groups: -NR5A1, combinations without NR5A1; +NR5A1, combinations with NR5A1 but without GATA4; +NR5A1 and GATA4, combinations with both NR5A1 and GATA4. (**B**) Quantitative data of EGFP+ cells in (**A**). n = 2, biological replicates, error bar indicates SD,

*Figure 3 continued on next page*

*Figure 3 continued*

three independent combination experiments were conducted. ~$10^4$ cells were analyzed in each experiment. (C) Immunofluorescent staining of KRT18 in 2F-hiSCs and dH1-2K7 after FACS. Scale bar = 100 μm. (D) Heat map of RNA-seq data illustrating differentially co-expressed genes. Red indicates upregulated expression, whereas green indicates downregulated expression. The genes were divided into three groups: Genes upregulated in 2F-hiSCs (dH1), 5F-hiSCs (dH1) and aSCs; genes upregulated in 5F-hiSCs (dH1) and aSCs but downregulated in 2F-hiSCs (dH1); genes downregulated in 2F-hiSCs (dH1), 5F-hiSCs (dH1) and aSCs; all compared to dH1-2K7. The gene number of each group was listed next to the map. Functional enrichment terms of each group and the representative genes were shown on the right side.

The online version of this article includes the following figure supplement(s) for figure 3:

**Figure supplement 1.** Heat map clustered by the expression of Sertoli cell markers in 2F-hiSCs, 5F-hiSCs and aSCs cells.
**Figure supplement 2.** PCA plot of differentially co-expressed genes in 2F-hiSCs (dH1), 5F-hiSCs (dH1) , 5F-hiSCs (HPF) and aSCs, compared to dH1-2K7.
**Figure supplement 3.** Immunofluorescent staining of SOX9 (red) in dH1, 2F-hiSCs, hESCs (H1) and adult Sertoli cells (aSCs).
**Figure supplement 4.** hiSCs and aSCs do not express Leydig cell specific markers.
**Figure supplement 5.** Induction of hiSCs from primary human skin fibroblasts.

development, immune effector processes and response to hormones. These genes, including *KRT18*, *PTGDS* and *SOX9*, are used as markers of Sertoli cells or are highly expressed in Sertoli cells (*Buganim et al., 2012*; *Sharpe et al., 2003*). We further compared the expression of 59 genes that are highly expressed in Sertoli cells (*Bouma et al., 2010*; *Boyer et al., 2004*; *Mincheva et al., 2018*) among the 2F-hiSCs, 5F-hiSCs and aSCs. All three groups exhibited similar expression of many markers that are expressed in more mature Sertoli cells, including *CDKN1B* (or *p27^{kip1}*) and *CLU* (*Wang et al., 2016*). However, some markers in the aSCs, including *NCAM2*, *INHA* and *KRT18* (*Kanatsu-Shinohara et al., 2012*; *Wang et al., 2016*), were expressed at lower levels (*Figure 3—figure supplement 1*) than those in the other two hiSCs. The Sertoli cell marker expression was very similar between the 2F-hiSCs and 5F-hiSCs. Principal component analysis (PCA) indicated that aSCs, 2F-hiSCs (dH1), 5F-hiSCs (dH1) clustered closely, confirming their co-expressed gene profiles are similar (*Figure 3—figure supplement 2*). SOX9 expressed at similar level in aSCs, dH1 and 2F-hiSCs but not in undifferentiated hESCs, showing that SOX9 expression increased after hESCs became fibroblasts, and maintained similar expression during 2F-hiSC formation (*Figure 3—figure supplement 3*). Therefore, we focused on the 2F-hiSCs for the subsequent more thorough characterization.

Although we used AMH:EGFP reporter to isolate Sertoli-like cells, it was still possible that the isolated cells were Leydig cells. Hence, we examined whether the cells expressed any Leydig cell marker 3β-HSD (*Zhang et al., 2015*). We confirmed that neither 2F-hiSCs nor aSCs expressed any 3β-HSD (*Figure 3—figure supplement 4A*). Based on the RNA-sequencing results, we also confirmed other gene markers of Leydig cells were not detected or expressed at low level (*Figure 3—figure supplement 4B*).

We also validated that NR5A1 and GATA4 was able to reprogram primary human skin fibroblasts (HSF) and yielded AMH:EGFP+ hiSCs (*Figure 3—figure supplement 5A, B and C*). 12.9% of AMH:EGFP+ cells were induced after overexpression of the 2TFs. KRT18 were only detected in the AMH:EGFP+ cells after induction along with other genes enriched in Sertoli cells (*Figure 3—figure supplement 5D and E*). These results further supported that 2TFs can reprogram other primary human cell types.

## 2F-hiSCs attract human endothelial cells and accumulate lipid droplets

Sertoli cells mediate the migration of endothelial cells to seminiferous tubules during testicular cord formation (*Brennan et al., 2003*; *Cools et al., 2011*). We investigated whether the hiSCs attracted human umbilical vein endothelial cells (HUVECs). The migration assay showed the number of HUVECs that were attracted and passed through the membrane was ~1.5 fold higher in both 2F-hiSCs and aSCs compared to the controls (*Figure 4A,B*). These results indicate that significantly more endothelial cells were attracted by the conditioned medium collected from the 2F-hiSCs than by that collected from the dH1-2K7 cells.

Another unique feature of Sertoli cells in humans and other mammalian species is the accumulation of lipid droplets in the cytoplasm (*Gorga et al., 2017*; *Wang et al., 2006*). The Gene Ontology analysis showed that genes that participate in lipid metabolism processes were upregulated in the

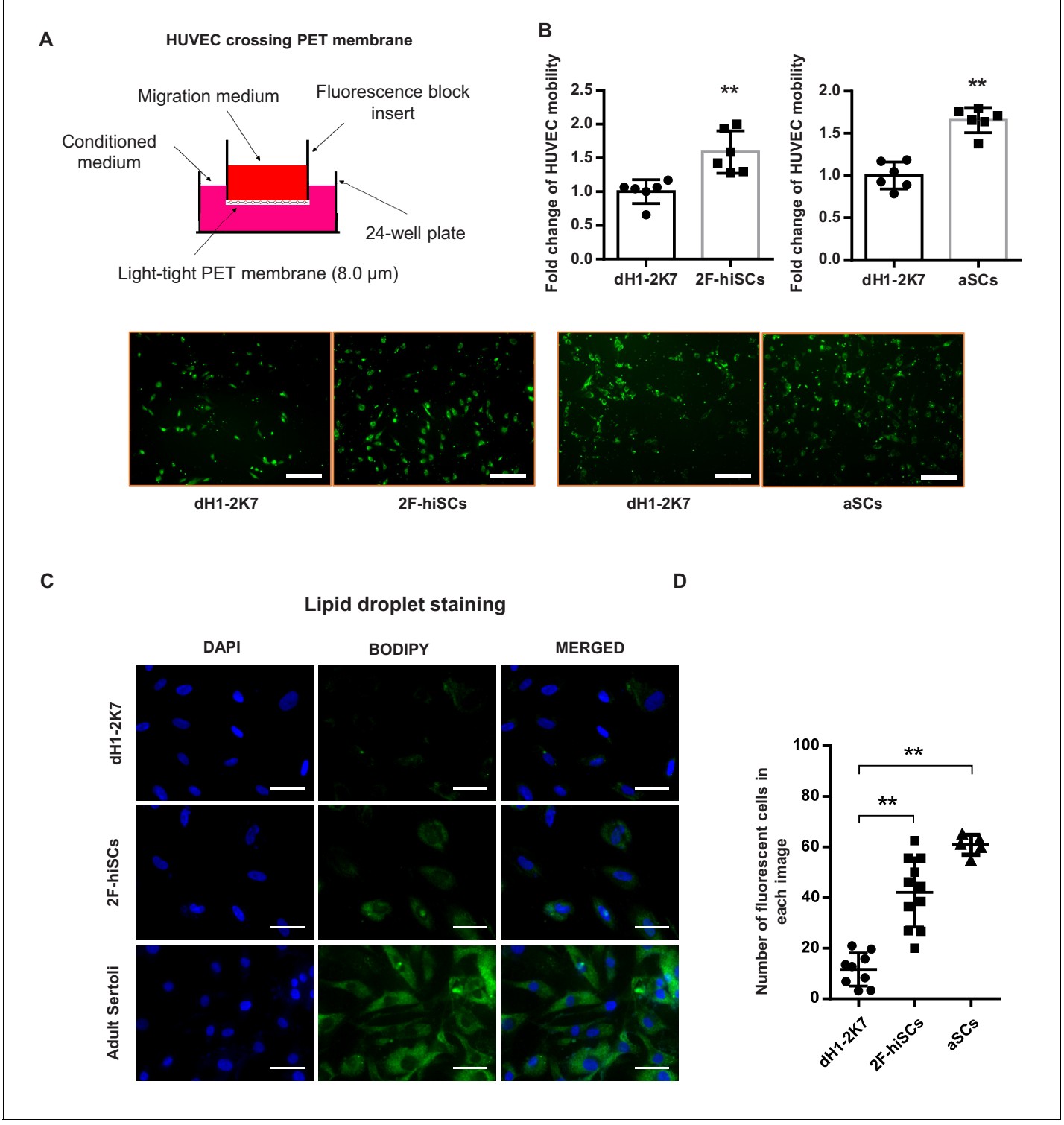

**Figure 4.** 2F-hiSCs attract endothelial cells and accumulate lipid droplets. (**A**) Mobility comparison of HUVECs incubated in conditioned medium collected from dH1-2K7, 2F-hiSCs and aSCs. Top, experimental diagram using the fluoroblok 24-multiwell insert plates with 8.0 μm pores. For observation, HUVEC were treated with 2.5 μM Calcein-AM fluorescent dye 1 hr prior to seeding. Down, fluorescent images showing cells passed through the pores. Conditioned medium collected from dH1-2K7 was used as control. Scar bar = 300 μm. (**B**) Summary of HUVEC cells crossing through the trans-well membrane. Each data point on the graph represents the number of cells in one separate image observed under 10 × fold microscope, area = 1690 × 1690 (μm). Error bars represent standard deviation of all the counted data points, n = 6, technical replicates, *p<0.05, **p<0.01, Student's t-test. two independent experiments were carried out. (**C**) BODIPY staining of lipid droplets in dH1-2K7, 2F-hiSCs and adult Sertoli

*Figure 4 continued on next page*

*Figure 4 continued*

cells (aSCs). All cells were fixed with 4% PFA and then stained with BODIPY for lipid droplets and DAPI for nucleus. Scar bar = 50 µm. (D) Each data point on the graph represents number of BODIPY+ cells in one separate image observed under 20 × fold microscope, area = 275 × 275 (µm). Error bars represent standard deviations of all the counted data points, *p<0.05, **p<0.01, Student's t-test. two independent experiments were carried out. The online version of this article includes the following figure supplement(s) for figure 4:

**Figure supplement 1.** Oil-red O staining showing lipid droplets (red) in dH1-2K7 (CTRL), aSCs and 2F-hiSCs.

2F-hiSCs, 5F-hiSCs and aSCs (*Figure 3D*), supporting the presence of high numbers of lipid droplets in the hiSCs. We used BODIPY to stain and count the number of cells with high numbers of lipid droplets to determine whether lipid droplets appeared in the 2F-hiSCs. The average percentage of cells that exhibited strong BODIPY positivity was approximately 15%, 40% and 60% in the control cells, 2F-hiSCs, and aSCs respectively (*Figure 4C,D*). Therefore, the percentage of cells containing high quantities of lipid droplets was higher in both 2F-hiSCs and aSCs than that in the dH1-2K7 cells. Oil-red O staining also confirmed the same result showing lipid droplet enrichment in 2F-hiSCs and aSCs (*Figure 4—figure supplement 1*).

## 2F-hiSCs sustain in vitro culturing of mouse spermatogonia cells

Mouse spermatogonia cells were isolated from seminiferous tubules of 6 dpp mice according to established protocols and co-cultured with dH1 or 2F-hiSCs to examine whether 2F-hiSCs sustain the growth of male germ cells (*Figure 5A*, *Figure 5—figure supplement 1A, B, C, D, E and F*). The observations that more mouse germ cells attached and survived on the 2F-hiSCs than dH1 cells began at approximately 12 hr of co-culturing (*Figure 5—figure supplement 1G*). The morphology of the round cells attached to the hiSCs appeared alive and resembled spermatogonia cells but the cells attached to dH1 appeared apoptotic and degenerated. After 48 hr of co-culturing, the samples were fixed and immunostained to further confirm that the 2F-hiSCs formed attachments with the spermatogonia cells. The immunostaining of the germ cell-specific marker DAZL (to identify mouse spermatogonia cells) and human-nuclear specific marker NuMA (to identify hiSCs and dH1) indicated that significantly more DAZL-positive spermatogonia cells attached to the hiSCs, but almost no DAZL-positive cells attached to the dH1 cells despite the similar numbers of plated hiSCs and dH1 cells (*Figure 5B and C*). Similar immunostaining patterns were observed when aSCs were co-cultured with mouse spermatogonia cells (*Figure 5—figure supplement 2*). Sertoli cells directly contact male germ cells in seminiferous tubules in vivo, and we hypothesized that the hiSCs would directly contact the spermatogonia cells. We immunostained the co-cultured cells with DAZL and KRT18 and found that many DAZL-positive cells localized to areas occupied by hiSCs, typically at the edge of the cell bodies (*Figure 5D*).

## 2F-hiSCs suppress T cell proliferation, IL-2 production and protect transplanted human cells

A specialized property of Sertoli cells is their functional role in creating an immune-privileged environment in seminiferous tubules to protect germ cells from immunological attacks. Previous studies have shown this unique function, which has been exploited in therapeutic transplantation for the protection of many other cell types (*Kaur et al., 2015*). We first investigated whether the medium from the 2F-hiSCs cells exhibited any suppressive effect on the proliferation of Jurkat E6 cells (human T lymphocytes) to examine whether the 2F-hiSCs could suppress the immunoreaction of immunological cells. The suppressive effect was evaluated using an assay commonly used to determine the active metabolism of a proliferating cell, that is, WST-1 (*Chui et al., 2011*; *Figure 6A*). The Jurkat E6 cells were treated with various concentrations of 2F-hiSCs-conditioned medium or aSCs-conditioned medium and exhibited a significant dose-responsive decrease in cell proliferation compared to that in cells treated with dH1-2K7-conditioned medium (*Figure 6B*). The proliferation level was ~35% lower in the Jurkat lymphocytes exposed to the highest concentration of the 2F-hiSC-conditioned medium or aSCs-conditioned medium. We collected the Jurkat cells and analyzed the protein levels of interleukin-2 (IL-2), which plays an essential role in the immune system. The ELISA indicated that the IL-2 levels were significantly lower in the cells cultured with the 2F-hiSCs-conditioned or aSCs-condictioned medium than those in the cells cultured with the dH1-2K7-conditioned medium

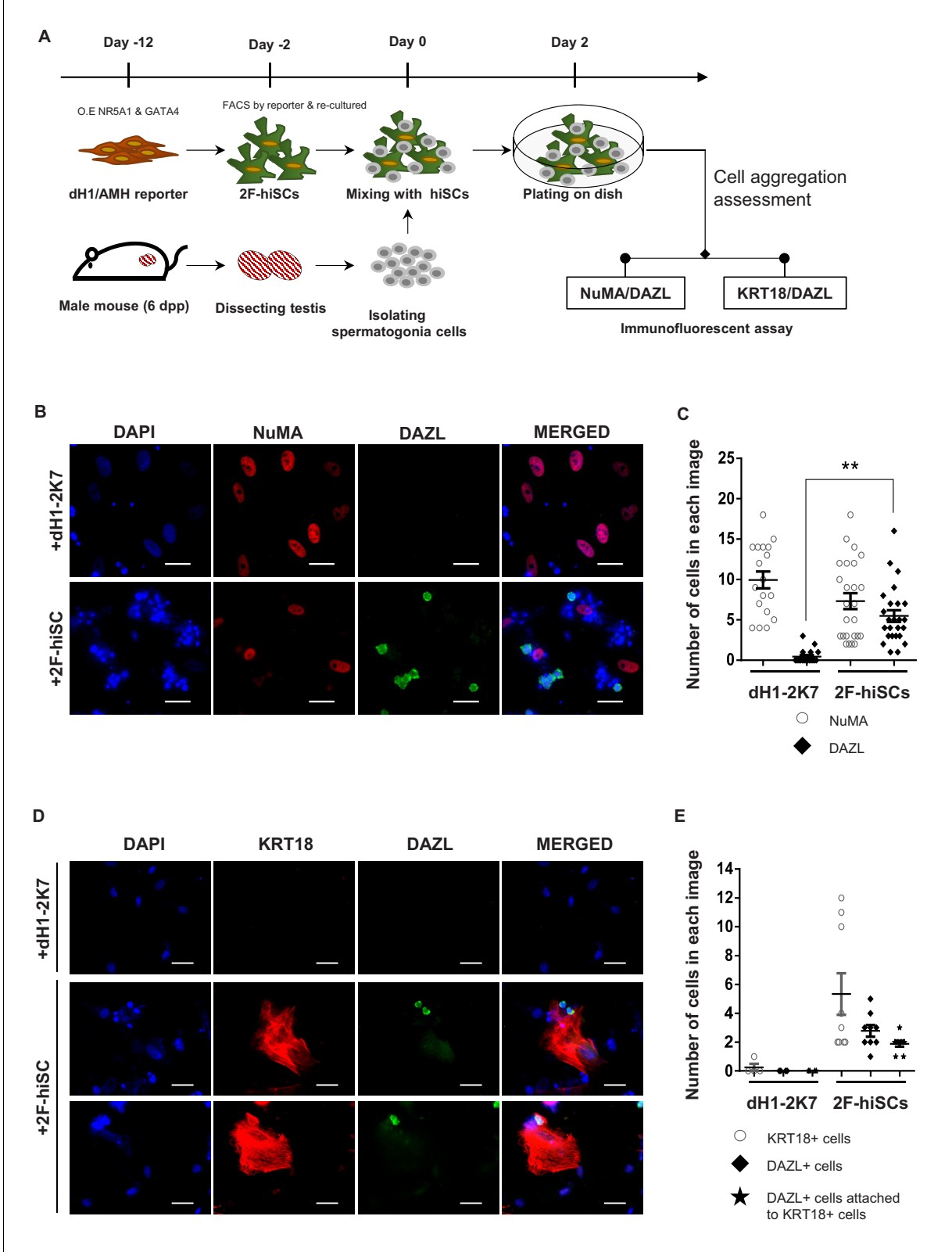

**Figure 5.** 2F-hiSCs sustain the viability of mouse spermatogonia cells. (**A**) Timeline and steps of co-culturing hiSCs with mouse germ cells. (**B**) Immunofluorescent staining of germ cell marker, DAZL (green), and human specific marker, NuMA (red), in co-cultured mouse spermatogonia cells with 2F-hiSCs or dH1-2K7 48 hr after plating. DAPI (blue) was used to indicate the nucleus. Scale bar = 50 μm. (**C**) Summary of cell numbers on the co-cultured plate in (**B**). Each data point indicates the number of cells counted in one separate image observed under 20 × fold microscope,

*Figure 5 continued on next page*

*Figure 5 continued*

area = 330 × 330 (µm). Error bars are the standard deviations of all the counted data points. dH1-2K7 (n = 18, technical replicates from two separated experiments), 2F-hiSCs (n = 25, technical replicates from two separated experiments). *p<0.05, **p<0.01, Student's t-test. (D) Immunofluorescent staining of DAZL (green) and KRT18 (red) in co-cultured mouse spermatogonia cells and 2F-hiSCs 48 hr after plating. DAPI (blue) was used to indicate the nucleus. Scale bar = 50 µm. (E) Summary of cell numbers on the co-cultured plate in (D). Each data point indicates the number of cell counted in one separate image observed under 20 × fold microscope, area = 282 × 330 (µm). Error bars are the standard deviations of all the counted data points. dH1-2K7 (n = 4, technical replicates from two separated experiments), 2F-hiSCs (n = 9, technical replicates from two separated experiments). *p<0.05, **p<0.01, Student's t-test.

The online version of this article includes the following figure supplement(s) for figure 5:

**Figure supplement 1.** Isolation of mouse spermatogonia cells and co-culturing with hiSCs.

**Figure supplement 2.** Immunofluorescent staining of germ cell marker, DAZL (green), and human specific marker, NuMA (red), in co-cultured mouse spermatogonia cells with aSCs or dH1-2K7 48 hr after plating.

(*Figure 6C*, figure 6-figure supplement 1C). 22 genes previously known to participate in immune-modulation (*Kaur et al., 2014*; *Mital et al., 2010*) or categorized as immune effector gene by gene ontology were activated after reprogramming of dH1 (*Figure 6D*), further supported that hiSCs acquired immunosuppressive function.

Approximately, 1.3 × 10^6 human 293FT cells stably integrated with a luciferase-expressing vector were co-transplanted with 2.5 × 10^5 dH1-2K7, 2F-hiSCs or aSCs into mice with normal immune systems via hypodermic injection to determine whether the 2F-hiSCs could protect xenotransplanted cells. The transplantation experiment was performed to investigate the immunosuppressive effects at different locations (foreleg, hindleg, left and right sides of the animals as indicated on the figures) in different animals, and the transplanted sites were monitored for up to 10 days. D-luciferin was injected into the animals to follow the surviving transplanted 293FT cells, and the signal was monitored using live imaging 15 min post-injection beginning 3 days after transplantation. The transplanted 293FT cells gradually diminished in the immunocompetent mouse from day 3 to day 10 (mouse #1, #2) as indicated by the reduced luciferase activity at the transplanted sites (*Figure 6E*, *Figure 6—figure supplement 1A*). All 293FT cells co-transplanted with hiSCs exhibited higher luciferase activity, which ranged from 1.7- to 3.9-fold, 3 days after transplantation. Three of the four groups of transplanted cells survived until day 10, and two of the three groups of 293FT cells with hiSCs survived at least 10 days after transplantation with strong luciferase activity. In contrast, their counterpart control cells exhibited less than 40-fold or no detectable luciferase activity (mouse #1 foreleg group and mouse #2 hindleg group). Similar level of protection was observed when the same number of 293FT cells were transplanted with aSCs into mouse #3 and #4 (*Figure 6—figure supplement 1B*). The transplants at day 10 were collected and dissected to examine the remaining cell types. NuMA staining indicated that there were many human cells survived in the transplants of 2F-hiSCs and some cells clustered together as cell aggregates (*Figure 6—figure supplement 2A*). These human cells could be 293FT cells or 2F-hiSCs. Positive immunostainings of NR5A1 confirmed the existence of 2F-hiSCs in the transplant but not the control transplant or the surrounding mouse tissue (*Figure 6—figure supplement 2B*).

## hiSCs forms cell aggregates and do not actively proliferate

Mouse Sertoli cells forms cell aggregates during in vitro culturing (*Buganim et al., 2012*). 2F-hiSCs also exhibited similar morphology of spherical cellular aggregates when cultured in 10% FBS on matrigel (*Figure 6—figure supplement 3A*). Different from the mouse Sertoli cells, we observed formation of partial ring-like structures when 2F-hiSCs were cultured in 2% FBS medium (*Figure 6—figure supplement 3B*). Adult human Sertoli cells tend to be more quiescent and less proliferative (*Sharpe et al., 2003*). We examined the proliferation of 2F-hiSCs by EDU staining and found that these cells were not proliferative (*Figure 6—figure supplement 4A and B*).

## CX43 deletion disrupts gap junctions and alters the expression profile of hiSCs

We investigated whether the deletion of the gap junction protein CX43 could affect hiSCs formation and determined whether hiSCs exhibit the same genetic requirements for development as Sertoli cells in vivo. We created a homozygous deletion of *CX43* hESC line, derived fibroblasts from this line

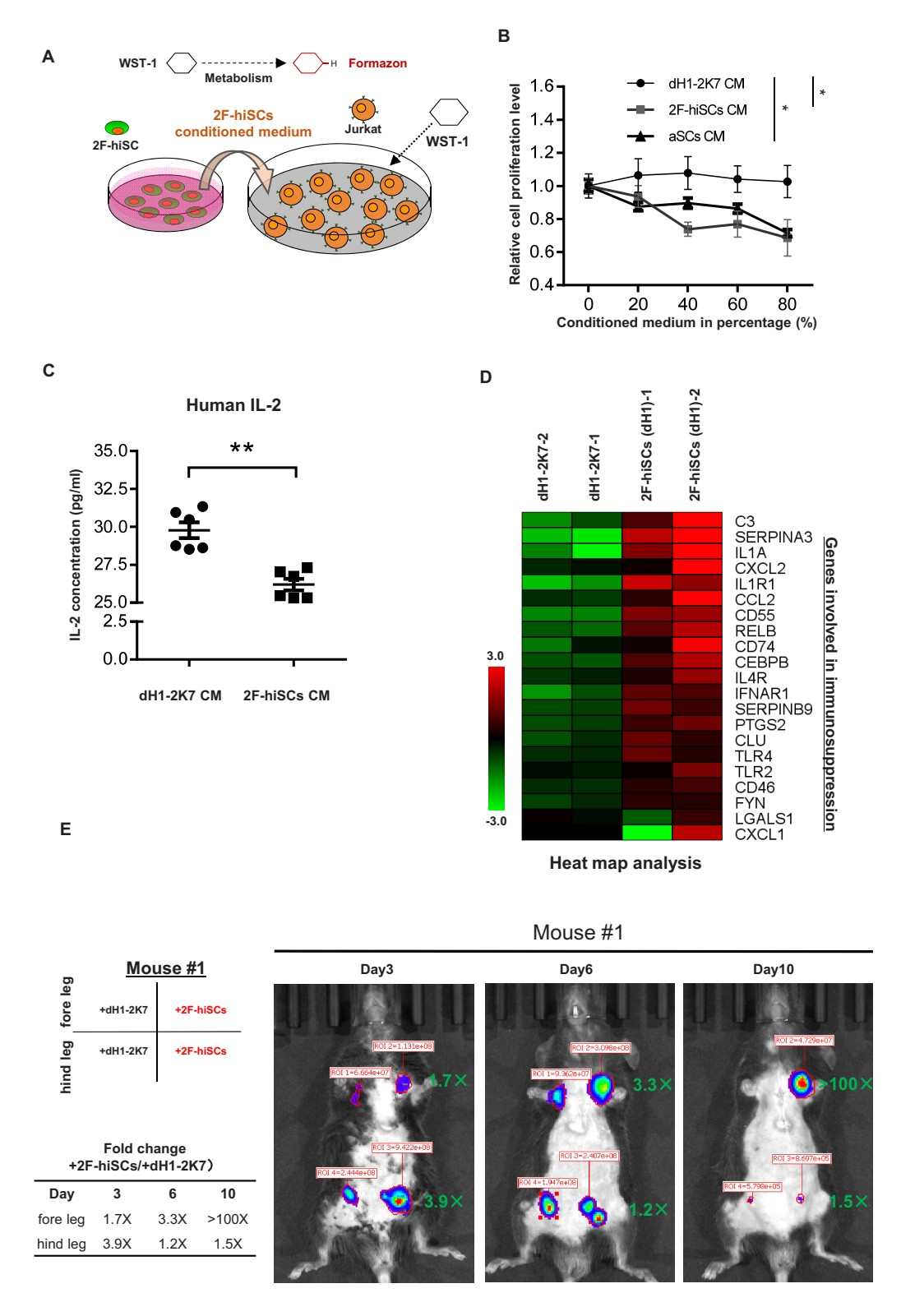

**Figure 6.** 2F-hiSCs exhibit immunosuppressive function and protect human cells in xenotransplantations. (**A**) Schematic illustration of the WST-1 assay to measure the inhibiting effect of 2F-hiSCs on Jurkat-E6 cell proliferation. Cleavage of the tetrazolium salt WST-1 by metabolically active cells results in the formation of formazon. The level of formazon is directly proportional to the proliferation level of the Jurkat cells. (**B**) Measurement of proliferation level in Jurkat cells treated with dH1-2K7, 2F-hiSCs or adult Sertoli cell (aSCs) conditioned medium. X axis represents the indicated proportion of

*Figure 6 continued on next page*

*Figure 6 continued*

conditioned medium tested in experiment. Conditioned medium from dH1-2K7 was used as a control. Values are presented as means ± SD and are from two separate experiments with each concentration tested in triplicate; Dunnett's test was carried out among samples at 80% CM, *p<0.05, **p<0.01. (C) Histogram showing the production of human IL-2 in lymphocytes treated by dH1-2K7 or 2F-hiSCs conditioned medium, as measured by ELISA. Error bars represent standard deviation of three technical duplicates from two separate experiments, *p<0.05, **p<0.01, Student's t-test. (D) A cluster of known immune-modulating genes for Sertoli cell immunoprotection. Relative gene expression level was indicated as red (upregulated) or green (downregulated). (E) Live imaging of luciferase-tracking assay 3 days to 10 days after transplantation. ~1.3 × 10$^6$ 293 FT cells were subcutaneously injected into the indicated sites of the mouse #1 with 2 × 10$^5$ dH1-2K7 or 2F-hiSCs cells. Cell type and location of transplantation are indicated at left. Numbers in red represent primary readings of luciferase activity. The normalized fold change of luciferase cell activity between hiSCs and dH1-2K7 in each time point was summarized in the table.

The online version of this article includes the following figure supplement(s) for figure 6:

**Figure supplement 1.** hiSCs protect the survival of human cells 293FT during xenotransplantation.

**Figure supplement 2.** Survival of human 293FT cells with hiSC in xenograft site.

**Figure supplement 3.** hiSCs form spherical cellular aggregates in culture medium.

**Figure supplement 4.** Proliferation assays showing hiSCs are not proliferative.

and compared the reprogramming efficiency of the hiSCs (*Figure 7A*). We successfully generated three targeted mutations at *CX43*, that is, one single allele mutation (6#) and two double allele mutations (21# and 26#) at Exon 2 of the *CX43* gene, using a CRISPR-CAS9 gene editing system (*Figure 7B*). The protein expression was completely disrupted in the CX43$^{-/-}$ (21#) and CX43$^{-/-}$ (26#) cell lines, but the heterozygous protein level remained similar to the wild-type CX43 level based on the Western blot analyses (*Figure 7—figure supplement 1A*). The immunostaining of CX43 (*Figure 7C*) and photo bleaching assay (*Figure 7D and E*) of ES-derived fibroblasts (dH1) both showed that the expression of CX43 and gap junctions between neighboring cells were disrupted because there was no detectable CX43 staining or diffusing fluorescent dye recovered after photo bleaching in the CX43$^{-/-}$ (26#) cell line.

We compared the reprogramming efficiency of the 2F-hiSCs between the wild-type dH1 and CX43KO (CX43$^{-/-}$ (#26)) cell lines. The time course experiments showed that the percentage of AMH:EGFP+ in the WT hiSCs peaked at ~13.8% on day 15 and decreased to 3.3% on day 25 (*Figure 7F and G*; *Figure 7—figure supplement 1B*). Remarkably, the percentage of AMH:EGFP+ in the CX43$^{-/-}$ (#26) cell line was 23.9% on day 15 and decreased to 18.5% on day 25, but the overexpression of CX43 in the deletion cell line revealed a much lower percentage of AMH:EGFP+ from days 4 to 25. Therefore, the expression level of CX43 in the cells was indirectly proportional to the percentage of AMH:EGFP+ cells. This result is consistent with the higher AMH expression in CX43 knockout mice reported in a previous study (*Weider et al., 2011*) and suggests that the effect of the CX43 deletion leads to the dedifferentiation of Sertoli cells to a less mature state.

The high percentage of AMH:EGFP+ cells in the CX43KO cells may be due to their more immature status than that of WT AMH:EGFP+ hiSCs. Therefore, we compared the transcriptional profiles of these two populations to examine whether CX43KO affected gene expression or any cellular processes. The volcano analysis revealed that 2736 genes were differentially expressed (*p*-value less than 0.01) between the CX43KO 2F-hiSCs and WT 2F-hiSCs (*Figure 7—figure supplement 2A*). We found 754 genes with a difference in the transcript level greater than two-fold; 512 genes were down-regulated and 242 genes were upregulated in the CX43KO 2F-hiSCs compared to those in the WT 2F-hiSCs (*Figure 7—figure supplement 2B*). We further analyzed the 754 genes using a heat map and GO analysis to identify specific genes or processes affected by CX43KO. The genes were classified into eight gene sets according to the expression patterns among the WT dH1, WT 2F-hiSCs, CX43KO dH1 and CX43KO 2F-hiSCs (*Figure 7H*, *Figure 7—figure supplement 2C*). The genes in Groups 1 and 4 exhibited lower expression levels in the WT 2F-hiSCs and CX43 2F-hiSCs than in the other two groups of control cells, suggesting that these genes were affected by the reprogramming process. The genes in Groups 5 and 6 exhibited similar patterns between the WT cells and CX43KO cells, suggesting that these two groups were affected by CX43KO. In contrast, the expression profiles of the genes in Groups 2, 3, 7, and eight in the WT and CX43KO hiSCs differed from those in their counterpart controls and between the WT and CX43KO hiSCs. These genes may reflect the cellular maturation status of these two reprogrammed cell populations. The GO analysis of enriched terms in these four groups revealed that Group two was enriched with genes

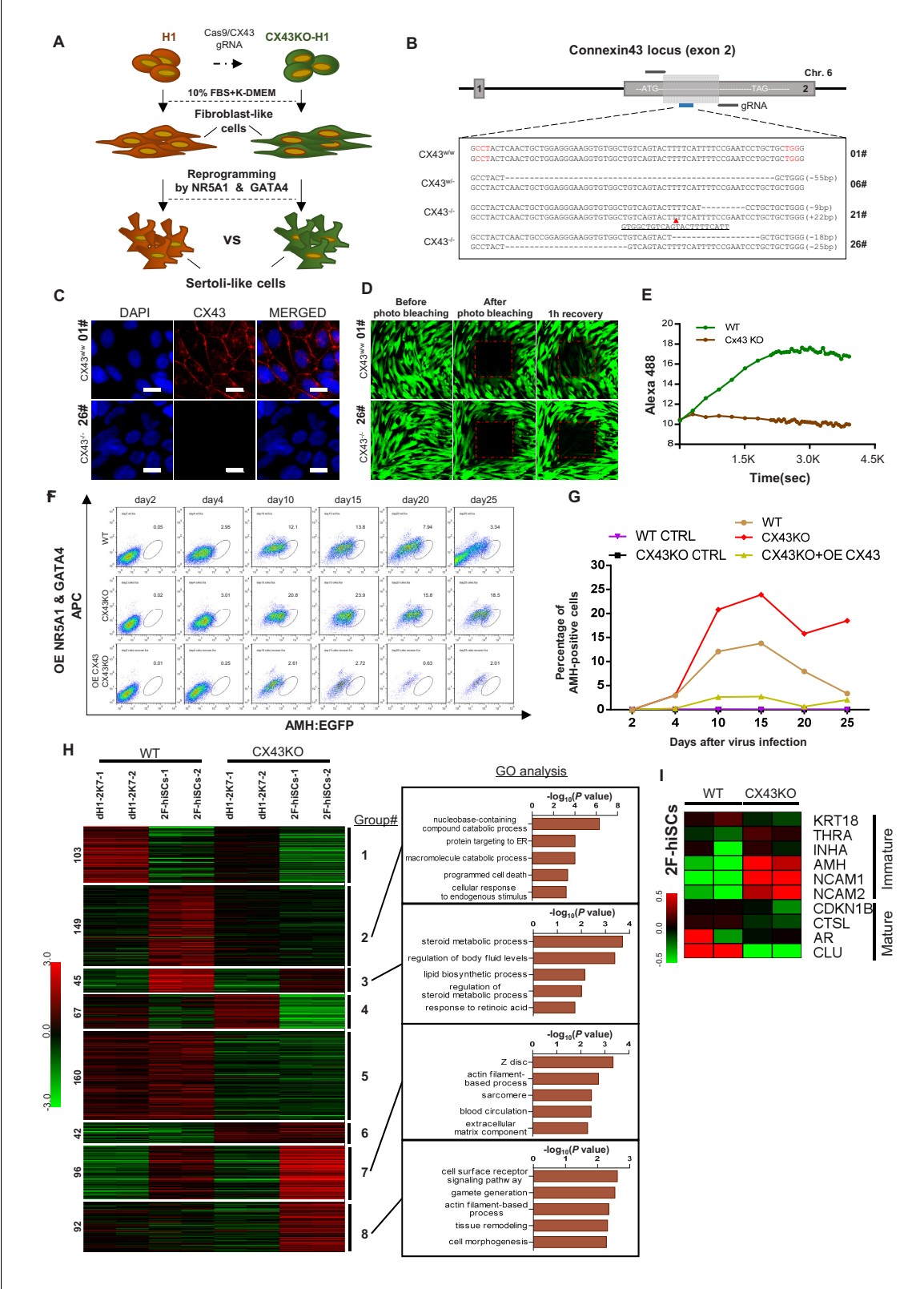

**Figure 7.** CX43 regulates maturation of Sertoli cells. (**A**) Schematic diagram showing the comparison of 2F-hiSCs cells reprogrammed from *CX43* knock out fibroblasts and wild type fibroblasts. (**B**) *CX43* knockout design and the targeted sequences of indicated hESC lines. The knockout region was located in the second exon of the *CX43* gene. Three knockout cell lines were correctly targeted and mutated: one was heterozygous (06#) and the other two were homozygous (21# and 26#). Dotted lines indicate deletion mutations, red triangles indicate insertion mutations. (**C**) Immunofluorescence

*Figure 7 continued on next page*

*Figure 7 continued*

analysis of CX43 (red) in wild type ES cell line (01#) and knock out cell line (26#). DAPI (blue) was used to indicate the nucleus. Scale bar = 20 μm. (**D**) Photo bleaching assay to test the Calcien AM transport ability of fibroblasts differentiated from wild type ES cell line (01#) and knock out cell lines (26#). (**E**) Measurement of Calcien AM signal over time after fluorescence bleaching. (**F**) Time course experiment during 2F-hiSCs reprogramming. The AMH: EGFP+ percentage resulted from three initial fibroblasts were tested: wild type dH1 (WT), CX43 knock out dH1 (CX43KO) and CX43 knock out plus CX43 overexpressed dH1 (OE CX43-CX43KO). (**G**) Summary of 2F-hiSCs reprogramming efficiency in (**F**). WT fibroblasts (WT) and CX43KO fibroblasts infected by p2k7 empty virus were also included. (**H**) Hierarchical clustering analysis using DEGs (FPKM value, fold change >2) in *Figure 7—figure supplement 2B*. Genes were classified into eight parts according to their expression pattern in CX43 knock out dH1, CX43KO 2F-hiSCs, wild type dH1 and wild type 2F-hiSCs. Gene Ontology analysis of genes in part 2, 3, 7, eight was shown on the right respectively. Other groups were shown in *Figure 7—figure supplement 2C*. (**I**) Heat map indicating expression level of immature and mature Sertoli cell markers in WT and CX43KO 2F-hiSCs. Relative gene expression level was indicated as red (upregulated) or green (downregulated).

The online version of this article includes the following figure supplement(s) for figure 7:

**Figure supplement 1.** Effect of CX43 knock out on protein level and percentage of AMH-GFP.

**Figure supplement 2.** The effect of CX43 knock out on transcriptome of AMH-GFP cells.

involved in catabolic processes, including nucleobase-containing compound catabolic process, and Group three contained genes involved in steroid metabolic or lipid biosynthetic processes. Group seven included many genes that participated in the cytoskeleton, and Group eight contained genes related to gamete generation.

Ten marker genes previously reported to be more highly expressed in mature or immature Sertoli cells (*Sharpe et al., 2003*; *Wang et al., 2016*) were chosen for the expression level comparisons between WT and CX43KO hiSCs. The CX43KO hiSCs exhibited a higher expression of some immature markers and lower expression of mature markers, suggesting that the CX43KO cells were more immature than the WT hiSCs.

## Discussion

This study reported that human fibroblasts can be reprogrammed to Sertoli-like cells using 5TFs (NR5A1, GATA4, WT1, DMRT1 and SOX9) or 2TFs (NR5A1 and GATA4). Fibroblasts from two sources, that is, human pulmonary fibroblasts and fibroblasts derived from human embryonic stem cells, were reprogrammed to Sertoli-like cells and exhibited transcriptomes similar to those of primary adult Sertoli cells. The present study generated a Sertoli cell-specific gene reporter, that is, AMH: EGFP, to enhance the efficiency of isolating a relatively pure population of hiSCs from a mixed population of cells at different reprogramming stages (*Figure 1*). AMH has been used in many conditioned knockout studies to specify the expression of Cre recombinase in mouse Sertoli cells (*Lécureuil et al., 2002*). Recent single-cell transcriptome study of adult human testicular cells indeed confirmed that AMH is specifically expressed in Sertoli cells but not other somatic cells or germ cells (*Wang et al., 2018*). We cloned the human promoter of *AMH* and fused it to EGFP and showed that the cell population expressing EGFP appeared only after reprogramming, and these cells expressed many Sertoli cell marker genes. This isolation step eliminates the possibility that the reprogrammed cells we studied were other somatic cells such as Leydig cells which share some gene markers. The success of reprogramming human fibroblasts with the same factors used in a previous mouse study (*Buganim et al., 2012*) confirms that the reprogramming capability of these 5TFs is conserved between humans and mice.

Another achievement of this study was the reduction in the reprogramming factors from 5TFs to 2TFs. NR5A1, which is often called SF1, was the most essential of the five factors because almost no AMH:EGFP+ cells appeared in the reprogramming experiments without NR5A1, even if GATA4, WT1, DMRT1 and SOX9 were overexpressed (*Figure 3A and B*). Notably, only the addition of GATA4, but not the addition of the three other TFs, to the reprogramming combination increased the percentage of AMH:EGFP+ cells. Several TFs, including NR5A1 and GATA4, are key regulators establishing the mouse Sertoli cell identity (*Rotgers et al., 2018*). Therefore, unsurprisingly, NR5A1 and GATA4 were sufficient to reprogram human fibroblasts to Sertoli-like cells. WT1 and SOX9 are also determining factors in mouse Sertoli cells, but their reprogramming abilities were lower than those of NR5A1 and GATA4. SOX9 interacts with NR5A1 to trigger the specific onset of AMH expression (*Rotgers et al., 2018*), but SOX9 overexpression in our reprogramming experiments only

slightly increased the percentage of AMH:EGFP+ cells in a few combinations of TFs (*Figure 3A and B*).

The cellular characterizations of the 2F-hiSCs showed that these cells carried many known properties of Sertoli cells. Lipid metabolism in Sertoli cells is important for providing nutritional and energy supplies to germ cells (*Gorga et al., 2017*). Previous studies have shown that the high metabolic activity requirements for lipid and steroid synthesis are associated with the differentiated status of Sertoli cells (*Johnston et al., 2008*). Notably, the hiSCs in the present study exhibited a high expression of lipid and steroid related genes (*Figure 3D*), and the WT hiSCs exhibited a higher expression than the CX43KO hiSCs (*Figure 7H*). These results suggest that the hiSCs were similar to terminally differentiated Sertoli cells in vivo. The lipid droplet staining assays clearly showed the presence of high numbers of lipid droplets in the hiSCs but not the fibroblasts used for reprogramming (*Figure 4C*).

Sustaining the viability and differentiation of spermatogonia cells is the most essential function of Sertoli cells. We examined the ability of hiSCs to sustain mouse spermatogonia cells because of the inaccessible procurement of human spermatogonia cells from hospitals. Previous studies performing xenotransplantation experiments transplanting primate spermatogonia stem cells into mouse testis demonstrated that primate germ cells colonized in the seminiferous tubules of the recipient mice, but the primate germ cells did not differentiate because of the evolutionary differences between these species (*Nagano et al., 2001*). The isolated mouse spermatogonia cells in our study attached and survived on the hiSCs, but no differentiated spermatocytes were detected (*Figure 5B*; data not shown). These results suggest that the reprogrammed Sertoli cells were only capable of providing a niche for the survival of male germ cells in vitro. Recent studies have reported in vitro derivations of immature human gametes from hESCs (*Easley et al., 2012*; *Jung et al., 2017*; *Kee et al., 2009*). The male and female gametes in these studies were immature likely due to the absence of fully competent somatic cells in the differentiating culture. Therefore, adding induced Sertoli cells or granulosa cells to in vitro differentiating cultures may be a key step in obtaining fully functional gametes in vitro.

Another prominent role of Sertoli cells is the creation of an immune-privileged environment to protect germ cells from immune attacks of lymphocytes. Many studies have documented that Sertoli cells secrete factors that suppress the proliferation of T cells, B cells and NK cells (*Kaur et al., 2014*; *Luca et al., 2018*), including the suppression of IL-2 from T cells (*Selawry et al., 1991*). Our results indicate that culture medium incubated with hiSCs suppress the proliferation of Jurkat cells (the immortalized cell line of human T lymphocytes) and reduce IL-2 production in Jurkat cells treated with hiSC-conditioned medium (*Figure 6B and C*). Many genes previously known to participate in immune effector processes were upregulated in the hiSCs compared to those in fibroblasts (*Figure 6D*), suggesting that hiSCs modulate immune suppression via multiple immunological pathways. Notably, all 293FT cells co-transplanted with hiSCs into two different immunocompetent mice survived longer than their counterpart control cells co-transplanted with fibroblasts (*Figure 6E* and *Figure 6—figure supplement 1A,B*). These results suggest that the hiSCs protected the xenotransplanted 293FT cells from host immune cells. Previous studies have reported that Sertoli cells protect many cell types in allogenic and xenogenic transplantation, and several studies used immunosuppressive drugs or immune-deficient mice in long-term follow-up experiments (*Kaur et al., 2015*; *Mital et al., 2010*). One study xenotransplanted porcine Sertoli cells with rat islet cells into the kidney capsule of immunocompetent rats (*Yin et al., 2009*). The islet allografts survived 8 to 9 days when $1.5 \times 10^6$ porcine Sertoli cells were co-transplanted. In our experiments, $2.5 \times 10^5$ hiSCs were co-transplanted with 293FT cells and survived at least 10 days in two mice. Therefore, our xenotransplanted 293FT cells survived a similar or longer duration with lower numbers of reprogrammed human Sertoli cells in immunocompetent mice. Long-term transplantations of hiSCs with clinically relevant cell types, including pancreatic islets and skin grafts, should be investigated for potential clinical applications, such as the treatment of diabetes and skin burns. Our hiSCs have the advantage of being of human origin, which alleviates the issue of xenotransplantation and potential of animal virus infection.

Reprogrammed Sertoli cells may be used as a model for examining the cellular and genetic mechanisms of human Sertoli cell biology. A previous study reported an association between Sertoli-only syndrome (SCO) and the lower mRNA expression of CX43 (*Defamie et al., 2003*) and suggested that the absence of CX43 rendered the Sertoli cells more immature. However, the precise

mechanism by which the absence of CX43 causes infertility is unclear. Our studies revealed that multiple pathways, including lipid metabolism and nucleobase catabolism, were more highly expressed in the WT than in the CX43KO hiSCs, indicating that the absence of CX43 disrupts these molecular pathways in Sertoli cells. The expression of several markers of immature Sertoli cells was higher and the expression of markers of mature Sertoli cells was lower in CX43KO than in the WT hiSCs. Taken together, our results suggest that the deletion of CX43 disrupts multiple molecular pathways and delays the maturation of Sertoli cells. Infertility in men may be caused by genetic mutations in germ cells or Sertoli cells. Genetic or cellular defects in Sertoli cells have not been well studied to understand how they cause infertility due to a lack of in vitro model of human Sertoli cells. Our in vitro induced system now provides a platform for basic research and potential treatment of male infertility caused by Sertoli cells in the future.

# Materials and methods

### Key resources table

| Reagent type (species) or resource | Designation | Source or reference | Identifiers | Additional information |
|---|---|---|---|---|
| Mouse | C57BL/6 | Vital River Laboratory Animal Technology | | |
| Cell line (*Homo-sapiens*) | 293FT cells | Thermo Fisher Scientific | Cat# R70007 | |
| Cell line (*Homo-sapiens*) | H1 ES cells | WiCell Research Institute | Cat# WA01 | |
| Cell line (*Homo-sapiens*) | Human Pulmonary Fibroblasts | National Infrastructure of Cell Line Resource | Cat# CCC-HPF-1 (PUMC, Beijing) | |
| Transfected construct (*Homo-sapiens*) | p2k7-EF1alpha-NR5A1 | This paper | | Lentiviral construct to express target gene |
| Transfected construct (*Homo-sapiens*) | p2k7-EF1alpha-GATA4 | This paper | | Lentiviral construct to express target gene |
| Transfected construct (*Homo-sapiens*) | p2k7-EF1alpha-SOX9 | This paper | | Lentiviral construct to express target gene |
| Transfected construct (*Homo-sapiens*) | p2k7-EF1alpha-WT1 | This paper | | Lentiviral construct to express target gene |
| Transfected construct (*Homo-sapiens*) | p2k7-EF1alpha-DMRT1 | This paper | | Lentiviral construct to express target gene |
| Transfected construct | p2k7-EF1alpha-luciferin | This paper | | Lentiviral construct to express target gene |
| Transfected construct (*Homo-sapiens*) | p2k7-EF1alpha-CX43 | This paper | | Lentiviral construct to express target gene |
| Recombinant DNA reagent | p2k7-AMH-GFP | This paper | | Lentiviral construct for AMH reporter |
| Recombinant DNA reagent | pX335-U6-Chimeric_BB-CBh-hSpCas9n(D10A) | Addgene #42335 | | Expressing Cas9 and gRNA (*Cong et al., 2013*) |
| Biological sample (*Homo-sapiens*) | Primary adult human Sertoli cells | Shanghai Jiao Tong University | | Freshly isolated |
| Antibody | anti-DAZL (Mouse polyclonal) | AbD Serotec | Cat#: MCA2336; RRID: AB_2292585 | IF(1:50) |
| Antibody | anti-NuMA (Rabbit polyclonal) | Abcam | Cat#: Ab97585; RRID: AB_GR27454-16 | IF(1:100) |
| Antibody | anti-human KRT18 (Rabbit polyclonal) | Proteintech | 10830–1-AP | IF(1:100) |
| Antibody | anti-SOX9 (Rabbit monoclonal) | Abcam | Cat#: Ab170660; RRID: AB_GR155689-1 | IF(1:100) |

*Continued on next page*

*Continued*

| Reagent type (species) or resource | Designation | Source or reference | Identifiers | Additional information |
|---|---|---|---|---|
| Antibody | anti-human CX43 (Rabbit polyclonal) | Cell Sigaling | Cat#: 3512S | IF(1:100) |
| Antibody | anti-human DMRT1 (Rabbit polyclonal) | Abcam | Cat#: Ab1786 | IF(1:100) |
| Antibody | anti-human WT1 (Rabbit polyclonal) | Proteintech | Cat#: 12609–1-AP | IF(1:100) |
| Antibody | anti-human GATA4 (Rabbit polyclonal) | Proteintech | Cat#: 19530–1-AP | IF(1:100) |
| Antibody | anti-human NR5A1 (Rabbit polyclonal) | Proteintech | Cat#: 18658–1-AP | IF(1:100) |
| Antibody | donkey anti-Mouse IgG (H+L) Highly Cross-Adsorbed Secondary Antibody, Alexa Fluor 488 | Invitrogen | Cat#: A-21202; RRID: AB_141607 | IF(1:1000) |
| Antibody | donkey anti-Mouse IgG (H+L) Highly Cross-Adsorbed Secondary Antibody, Alexa Fluor 555 | Invitrogen | Cat#: A-31572; RRID: AB_1567203 | IF(1:1000) |
| Strain, strain background (*Escherichia coli*) | TransStbl3 Chemically Competent Cell | TransGen Biotech | Cat#: CD521 | |
| Peptide, recombinant protein | bFGF, Recombinant Human FGF basic Protein | R and D Systems | Cat#: 233-FB-CF | |
| Chemical compound, drug | G418, Geneticin | Thermo Fisher Scientific | Cat#: 10131035 | |
| Chemical compound, drug | Blasticidin | Thermo Fisher Scientific | Cat#: R21002 | |
| Software, algorithm, website | ImageJ | NIH | | https://imagej.nih.gov/ij/ |
| Software, algorithm, website | FlowJo (v10.3) | BD | | https://www.flowjo.com |
| Software, algorithm, website | Prism7 (v7.0 a) | Graphpad Software | | https://www.graphpad.com/scientific-software/prism/ |
| Software, algorithm | DAVID Bioinformatics Resources (v6.8; GO) | | | https://david.ncifcrf.gov |
| Software, algorithm, website | Tophat2/cufflinks | (*Kim et al., 2013*) | | http://ccb.jhu.edu/software/tophat |
| Software, algorithm, website | R (v3.4.1; PCA, cluster and DEG) | | | https://www.R-project.org |
| Other | DAPI stain | Invitrogen | D1306 | (1 µg/mL) |

## Experimental model and subject details

### Animal Care and Use

C57BL/6 mice were purchased from Vital River Laboratory Animal Technology Co., Ltd (Beijing, China). All animal maintenance and experimental procedures were performed according to the guidelines of the Institutional Animal Care and Use Committee (IACUC) of Tsinghua University, Beijing, China.

### Cell Lines

Human ES cell line used in this study were H1 (XY), purchased from WiCell, Inc Undifferentiated H1 were maintained on MEF feeder cells as previous described (*Jung et al., 2017*). All cells were

cultured at 37°C in a humidified incubator supplied with 5% $CO_2$. ES medium were standard knock-out serum replacer (KSR) consisted of 20% knockout serum replacer, 0.1 mM nonessential amino acids, 1 mM L-glutamine, 0.1 mM -mercaptoethanol, and 4 ng/ml recombinant human basic fibroblast growth factor (bFGF, R and D systems).

To obtain adherent fibroblast differentiation, human ESC clones were first transferred to 1% Matrigel coated plates by colony picking using a glass needle and cultured in ES cell conditioned medium (ES medium incubated overnight on irradiated MEFs) for ~5 days to remove all the residual MEF cells. Differentiation of hESCs to fibroblasts (dH1) began after aspirating of conditioned medium, washing with PBS without $Ca^{2+}$ and $Mg^{2+}$ twice, and replacing with differentiation medium (knockout DMEM with 20% fetal bovine serum, 0.1 mM nonessential amino acids, 1 mM L-glutamine). Differentiation medium was changed every 3 days. When the cell reached 90% confluency (~7 days), the resulting dH1 were collected and kept in liquid nitrogen tank for long time preservation or passaged by the ratio 1:3 if more cells were needed.

Primary human pulmonary fibroblasts (HPF) were purchased from National Infrastructure of Cell Line Resource. Primary human skin fibroblast (HSF) was a gift from Professor Ting Chen at National Institute of Biological Sciences, Beijing, and prepared according to the published protocol (*Qian et al., 2018*). Human adult Sertoli cells were a generous gift from Professor Zheng Li at Shanghai Jiao Tong University and prepared according to the established protocol (*Wen et al., 2017*). The HPF and HSF cells were maintained at 75 ml culture flask in 15 ml DMEM culture medium (Corning) containing 10% FBS (Gibco), 0.1 mM nonessential amino acids and 1 mM L-glutamine. Primary adult Sertoli cells were cultured as previously described in DF12 medium consisting of DMEM/F12% and 10% FBS (Gibco) and passaged every 5 days when cell confluence reached 80%.

Human umbilical vein endothelial cells (HUVEC) were a gift from Professor Jie Na at Tsinghua University. HUVECs were cultured in ECM medium (Sciencell) containing 10% FBS (Gibco) and passaged every 3 days when cell confluence reached 80%.

All of the cells above were cultured in a 37°C humidified incubator supplied with 5% $CO_2$.

## Method details

### Overexpression vector construction and lentivirus production

All overexpression vectors carrying EF1α promoter and desired gene were constructed using the Gateway system (Invitrogen) as previously described (*Jung et al., 2017*). Briefly, the candidate cDNA was first introduced into pENTR/1A or pENTR/D-topo donor vectors and transferred to 2K7 destination vectors with EF1α promoter in pENTR/5′ topo by LR recombination (*Suter et al., 2006*). Modified destination plasmids containing the cDNA were then introduced into 293FT cells together with the helper plasmids vsvg and Δ8.9 by LIPO3000 (Invitrogen) transfection to produce virus. Approximately, a total of 37 ml of virus supernatant was harvested on day1 and day3 after transfection and filtered with a 0.45 μm filter. At the time of virus infection, 8 μg/ml of polybrene was supplemented to increase infection efficiency.

### AMH:EGFP reporter and creating reporter cell line

1.6 kb of human *AMH* promoter upstream of the transcriptional start site was PCR amplified from genomic DNA of 293FT cells and cloned to pENTR5′-TOPO. Cloned plasmids were then recombined with pENTR/D-TOPO that carried the EGFP cDNA to create p2K7-AMH:EGFP recombinant plasmid and generated lentiviral supernatant as described above in overexpression vector construction section. Fibroblast HPF or dH1 in early passage (with 50% confluence) were transduced overnight on plate in fibroblast medium and recovered for one day after removal of virus. Subsequent drug selection by blasticidin (10 μg/ml) required another 3 days. Selected human fibroblasts were passaged for two times to expand the cell number for further experiments or frozen in liquid nitrogen.

### hiSC reprogramming and enrichment by FACS

Sertoli-like cell reprogramming was carried out in a T75 flask coated with Matrigel (1%). In brief, $1.5 \times 10^5$ human fibroblast cells carrying AMH:GFP reporter were seeded into T75 flask 1 day prior to overnight transduction with lentivirus NR5A1, GATA4, DMRT1, SOX9 and WT1 (for 5F-hiSCs) or NR5A1 and GATA4 (for 2F-hiSCs), recovered for 24 hr, following by drug selection with geneticin (1 mg/ml) for 5 days in MEF medium. After drug selection, transduced cells were cultured in DF12

medium, maintained for the indicated reprogramming duration, and harvested for FACS by digestion with TrypLE Express (Invitrogen). The single cell suspension for FACS was prepared with MACS medium (10% FBS in PBS with 0.0125 mM EDTA) and filtrated through a BD cell sorter. Cell sorting was proceeded on a high-speed cell sorter (Influx, BD) and was sorted to collecting tube containing DF12 medium.

Experiment aiming at examining different combinations of reprogramming factors was performed in six well plates and all possible combinations of NR5A1, GATA4, DMRT1, SOX9 and WT1 were tested. At the time of virus infection, 200 ul of each virus was added per well, and brought to the same final volume of 1 ml with p2k7 empty virus.

## Quantitative PCR and statistical analysis

Quantitative PCR was conducted as previously described (*Jung et al., 2017*). Briefly, total RNA was collected according to the instructions provided by QIAGEN RNeasy kit (QIAGEN) or TRIZOL (Invitrogen). CDNA was generated by EasyScript One-Step gDNA Removal and cDNA Synthesis Super-Mix kit (TRANSGEN) according to manufacturer's protocols using up to 500 ng RNA for each sample. 20 µl reaction (for Bio-rad 96-well System) were prepared and conducted with TransStart Green qPCR SuperMix kit (TRANSGEN). Gene expression was calculated using Bio-Rad CFX Manager program for relative expression formulation (dC(t)) and normalized to housekeeping genes (ACTB or GAPDH). Then, the gene expression of different samples were again normalized to the expression of control cells infected by p2k7 empty virus (CTRL) and reported as fold change (*Vandesompele et al., 2002*). Statistical analysis was carried out using Student's t-test or one-way ANOVA by Prism 6.0 software.

## RNA sequencing

~$2.5 \times 10^5$ cells were collected by FACS and total RNA was extracted by TRIZOL (Invitrogen). The quality and integrity of the purified RNAs were checked by Agilent 2100 bioanalyzer. Qualified RNA from the following samples was used for RNA Sequencing analysis: (1) HPF or dH1 carrying AMH: EGFP reporter, transduced with empty virus p2k7 and followed the same reprogramming procedure as described above. (2) Day10 hiSCs generated with five factors (5F-hiSCs) or generated with two factors (2F-hiSCs). (3) Human primary adult Sertoli cells (aSCs) cultured in vitro in DMEMF12 medium + 10% FBS. Sequencing libraries preparation and sequencing operations were carried out by ANNO-ROAD, a company providing RNA sequencing service, complied with the whole set of processes from Illumina.

## Immunofluorescence of cultured cells

Dissociated cells from FACS enrichment were collected onto a slide by Cytospin (800 r.p.m. for 5 min) or replated onto a 6-well plate. After that, cells were washed one time with PBS, fixed in 4% paraformaldehyde for 10 min and treated with 2.5% Triton X-100 for 15 min. For antibodies staining, slides were first blocked in 2.5% donkey serum for 1 hr, then incubated overnight in 4°C with primary antibody (1:200 for KRT18, 1:100 for NuMA, 1:200 for CX43, 1:100 for VASA, all rabbit-derived, Abcam; 1:50 for DAZL, mouse-derived, AbD Serotec; 1:100 for NR5A1, GATA4, WT1, SOX9, DMRT1, all rabbit-derived, Proteintech). Slides were then washed five times (each 3 min) with PBST (0.1% Tween-20/PBS), followed by anti-rabbit secondary antibodies (anti-rabbit-555 or anti-rabbit-488, Invitrogen) incubating for 1 hr at room temperature and washing for another five times. All sections were then mounted with Prolong Diamond Anti-Fade Mounting Reagent (ThermoFisher) and cover slip.

## Effect of 2F-hiSCs conditioned medium on Jurkat cell proliferation and IL-2 production

1 ml conditioned medium were collected from $3.5 \times 10^4$ of 2F-hiSCs or dH1-2K7 fibroblast (control) cultured at 50% confluence 48 hr after plating. For each assay, $2 \times 10^4$ human T lymphocytes (Jurkat E6-1 cells, gift from Professor Hai Qi, Tsinghua University) were seeded in 1 well of 96-well plate with 120 µl 1064 medium with 10% FBS, 0.1 mM nonessential amino acids, 1 mM L-glutamine, 0.1 mM–mercaptoethanol and added with indicated amount of conditioned medium from either 2F-hiSCs or dH1-2K7 (as control). The Jurkat cells were then cultured in a 37°C incubator with 5% $CO_2$

for 3 days. Metabolism of WST-1 was used to determine the proliferative ability of lymphocytes in each well according to manufacturer's instructions (Beyotime, Co., Ltd.). Three hours prior to analysis, 12 µl of WST-1 was added to each well and the final absorbance was measured by a microplate reader at 450 nm (SPECTRA max PLUS, Molecular Devices Inc). 1064 medium alone with WST-1 was used as a control to subtract background absorbance. For IL-2 quantification, $2 \times 10^5$ of Jurkat cells were seeded in one 6-well plate containing either 50% 2F-hiSCs or dH1-2K7 conditioned medium and cultured for 3 days. Then, cells were collected and lysed with 20 µl RIPA buffer. The concentration of IL-2 was determined by ELISA kit according to the manufacturer's instructions (Cusabio Biotech, Co., Ltd.).

## Cell migration assays

Confluent P8 HUVEC cells were cultured with fresh ECM medium overnight before experiment. Then cells were incubated with medium containing 2.5 µM Calcein-AM fluorescent dye for one hour in incubator supplied with 37°C, 5% $CO_2$. Cells were then trypsinized, counted and suspended in migration medium (50% ECM+50% MEF medium). Migration assays were carried out in Corning FluoroBlok 24-multiwell insert plate with 8.0 µm pores (Cat. No. 351157. Corning). Prior to seeding HUVEC (150 k, in 100 µl volume) into the insert, 50% conditioned medium pre-incubated with hiSCs or dH1 was mixed with 50% ECM and added to the basal chamber (600 µl in total). Following incubation of HUVECs at 37°C, 5% CO2 needed for 20 hr. The migration cells passed through the membrane were monitored by Calcein-AM with the help of an inverted microscope (Leica) at 485/535 nm (Ex/Em). Images were captured using Leica-Pro software provided by the microscope company. The number of migration cell was measured by calculating of the Calcein-AM green signal of each image.

## Mouse spermatogonia isolation and coculture with 2F-hiSCs

Mouse spermatogonia cells were isolated from ~24 testis of C57BL/6 mice at day six after birth according to previous protocol with minor modifications (*Kanatsu-Shinohara et al., 2003*; *Wang et al., 2015*). Briefly, the decapsulated testis and seminiferous tubules were detached with the help of tweezers. After washing with DPBS for three times, the seminiferous tubules were transferred to a new 15 ml tube and subjected to enzymatic digestion. Firstly, the seminiferous tubules were digested with 1 mg/ml collagenase IV for 5 min at 37°C in a water bath with shaking, then centrifuged at $50 \times g$ to collect the segregated tubules and washed three times with DMEM/F12 medium. The small fragments were resuspended in 2 ml culture medium composed of DMEM/F12 medium with 10% FBS and seeded on 2 wells of 6-well plates (Corning) and incubated at incubator supplied with 37°C, 5% CO2. After attachment for 24 hr, the somatic cells were tightly attached to the dish and formed patches, while spermatogonia were just loosely attached on the somatic cells. The spermatogonia were then transferred to a new plate coated with 1% matrigel and cultured for two passages in the medium (DMEM/F12 medium, 20% KSR, $100 \times$ Glutamax, $100 \times$ NEAA, $100 \times$ Pen/Strep, human GDNF 20 ng/mL, human bFGF 5 ng/mL). 24 hr prior to co-culturing experiments,~$1.5 \times 10^5$ dH1 or 2F-hiSCs were plated to 1 well of the 48 well plate. The spermatogonia cells were collected by gently washing with fresh medium and transferred to the well plated with the dH1 or 2F-hiSCs in DMEM/F12 medium with 10% FBS.

## Co-transplantation of hiSCs with 293FT cells

All animal experiments were approved by the Institutional Animal Care and Use Committees at Tsinghua University. ~ $1.3 \times 10^6$ number of 293FT cells stably transduced with a luciferase reporter were mixed with $2.5 \times 10^5$ cells of either dH1 or 2F-hiSCs in 100 µl Matrigel. The suspension was transplanted into C57BL/6 (Purchased from Vital River Laboratory Animal Technology) mice with normal immune system by subcutaneous injection. Live imaging of transplanted mice was performed at the indicative day after transplantation. 15 min prior to live imaging, 100 µl of D-luciferin (15 mg/ml, 10 µL/g mouse weight) was injected to the mice by intraperitoneal injection and luciferase activity was measured in the IVIS Spectrum machine (PerkinElmer Health Sciences, Inc).

## BODIPY and Oil Red O staining of lipid droplets

Cells were washed with PBS and fixed with 4% PFA for 15 min. For BODIPY staining, cells were stained with BODIPY (Invitrogen, working solution1 µg/ml) at room temperature for 15 min and washed in PBS for three times. Then, mounted cells with Prolong gold anti-fade reagent (Invitrogen).

For Oil Red O staining, 0.5% Oil Red O Stock Solution (in isopropanol) was used. Staining solution contains 6 parts of stock Oil Red O and 4 parts of distilled water, filtered with Whitman paper before used. After cell fixation, stained the cells with 1 ml staining solution for 15 min, then clear background using 60% isopropanol. Finally, wash with distilled water.

## Cell lines

All primary cells and cell lines have been tested to confirm no mycoplasma contamination before they were used in the experiments. Information about three established cell lines 293FT, H1 and CCC-HPF-1 were listed in the key resource table. None of the cell line is found in the list of misidentified cell lines maintained by the International Cell Line Authentication Committee.

## Quantification and statistical analysis

Statistical details of analysis including statistical test used, value of n and statistical significance were all described in the figure legends.

## RNA-seq data processing

All RNA-seq data were mapped to human genome build hg19 (UCSC) by TopHat (version 2.1.1), reads from PCR duplicates were dropped. The gene expression level was calculated by Cufflinks (version 2.2.1) using the protein coding genes GTF file extracted from Ensembl database (Homo_sapiens.GRCh37.75.gtf). Read counts were obtained using HTSeq (version 0.9.1). Differentially expressed genes (DEGs) were analyzed using R package DESeq2 and selected using p-value<0.01 as a threshold. Heat map was plotted using heatmap.2 function of R, and gene expressions were scaled to FPKM or Z-scores. We used R function cor to do sample correlation clustering and principal component analysis (PCA). K-means clustering was performed using Cluster 3.0 package (K = 3, Spearman Correlation, Complete-linkage) and clustered heat maps were produced by TreeView.

## Acknowledgements

We are indebted to Prof. Zuping He helping us to obtain primary human Sertoli cells, Prof. Ting Chen for providing us the primary human keratinocytes, and Prof. Jianbin Wang for helping us to analyze CNV of Sertoli cells. Research funding is provided by the Ministry of Science and Technology of China [2018YFA0107703, 2017YFC1001601] and Cross-discipline Foundation of Tsinghua University.

## Additional information

### Funding

| Funder | Grant reference number | Author |
| --- | --- | --- |
| Ministry of Science and Technology of the People's Republic of China | 2018YFA0107703 | Kehkooi Kee |
| Ministry of Science and Technology of the People's Republic of China | 2017YFC1001601 | Kehkooi Kee |

The funders had no role in study design, data collection and interpretation, or the decision to submit the work for publication.

### Author contributions

Jianlin Liang, Conceptualization, Formal analysis, Validation, Investigation, Writing—original draft; Nan Wang, Formal analysis, Validation, Investigation; Jing He, Data curation, Formal analysis,

Validation; Jian Du, Yahui Guo, Formal analysis, Validation, Methodology; Lin Li, Supervision, Validation, Methodology; Wenbo Wu, Chencheng Yao, Validation, Investigation, Methodology; Zheng Li, Resources, Validation; Kehkooi Kee, Conceptualization, Resources, Data curation, Formal analysis, Supervision, Funding acquisition, Validation, Investigation, Visualization, Methodology, Writing—original draft, Project administration, Writing—review and editing

### Author ORCIDs
Jianlin Liang ⬛ https://orcid.org/0000-0003-3027-2421
Kehkooi Kee ⬛ https://orcid.org/0000-0001-6926-7203

### Ethics
Animal experimentation: C57BL/6 mice were purchased from Vital River Laboratory Animal Technology Co., Ltd (Beijing, China). All animal maintenance and experimental procedures were performed according to the guidelines of the Institutional Animal Care and Use Committee (IACUC) of Tsinghua University, Beijing, China (Approval number: 17-JJK1).

### Decision letter and Author response
Decision letter https://doi.org/10.7554/eLife.48767.sa1
Author response https://doi.org/10.7554/eLife.48767.sa2

## Additional files

### Supplementary files
• Transparent reporting form

### Data availability
All data generated or analysed during this study are included in the manuscript and supporting files.

The following dataset was generated:

| Author(s) | Year | Dataset title | Dataset URL | Database and Identifier |
|---|---|---|---|---|
| Liang J | 2018 | HiSC RNA sequence data | https://www.ncbi.nlm.nih.gov/geo/query/acc.cgi?acc=GSE133757 | NCBI Gene Expression Omnibus, GSE133757 |

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
