## [Decision Letter]

**Acceptance summary:**

Your study nicely translates previous studies on reprogramming to the Sertoli lineage, provides a nice tool to model fertility syndromes and could potentially be utilized for clinical efforts aimed at fertility issues.

**Decision letter after peer review:**

Thank you for submitting your article "Induction of Sertoli-like cells from human fibroblasts by NR5A1 and GATA4" for consideration by *eLife*. Your article has been reviewed by three peer reviewers, and the evaluation has been overseen by a Reviewing Editor and Utpal Banerjee as the Senior Editor. The reviewers have opted to remain anonymous.

The reviewers have discussed the reviews with one another and the Reviewing Editor has drafted this decision to help you prepare a revised submission.

Summary:

The reviewers and this editor generally agree that this manuscript represents an important advance in the generation of human Sertoli cells, and goes beyond previously published work in mouse models. However, the reviewers raised several important issues that, if addressed, could potentially make the study acceptable for *eLife*.

Essential revisions:

All of the reviewer comments are relevant, and none appeared to raise issues that are beyond the scope of the study. We hope that you can carefully and thoughtfully respond to these reviews, with new data, where applicable.

Reviewer #1:

This study establishes methodology for direct reprogramming of human fibroblasts in Sertoli-like cells via expression of transcription factors expressed in fetal Sertoli cells. Given the dependence of spermatogonia upon somatic support cells, deriving human Sertoli cells is an important step toward the goal of producing mature gametes in vitro.

This is a thorough and careful study in which the authors optimize the number and combination of factors for reprogramming, show that fibroblasts from two different sources can be reprogrammed, and interrogate gene expression and function in the cells they derive. Functional tests include chemoattraction of endothelial cells, suppression of T cell proliferation, accumulation of lipid droplets, and capacity to support mouse spermatogonial survival. The most definitive test of function for in vitro-derived Sertoli cells would be their demonstrated support of complete spermatogenesis, however a barrier to this revealed by previous work of others is the failure of primate spermatogonia to complete maturation in the mouse testis. The most reasonable alternative is a transplant into non-human primate testes, which lies beyond the scope of the current study. Finally, the demonstration that connexin 43 deficient fibroblasts cannot produce Sertoli-like cells with equivalent function adds to rigor of the work. The results and approaches are well described in the figures and with cartoons.

Reviewer #2:

In this study, human fibroblast was first reprogrammed into Sertoli-like cells with five transcriptional factors and a gene reporter carrying the AMH promoter. they further reduce the number of reprogramming factors to two, NR5A1 and GATA4, and show that the hiSCs have transcriptome profiles and cellular properties that are similar to those of primary human Sertoli cells. The function of hiSCs was further tested by sustain of mouse spermatogonia cells, suppress the proliferation of human T lymphocytes and protect xenotransplanted human cells in mice. They also find that 2F-hiSCs attract human endothelial cells and accumulate lipid droplets. This is the first time induce human fibroblast to transform into Sertoli-like cells, the results are potentially interesting. The in vitro induced system provides a platform for basic research and potential treatment of male infertility in the future. However, there are several issues need to be addressed.

1) In this study, AMH was used as a reporter to identify the hiSCs. It has been reported that AMH is abundantly expressed in mouse Sertoli cells during embryonic stage and AMH protein is not detected in Sertoli cells after birth. The expression of AMH is probably different from other species. It is necessary to examine the expression of AMH in human Sertoli cells by immunostaining. The specificity of AMH promoter also need to be verified by transfecting human adult Sertoli cells.

2) The authors claim that human fibroblasts could be transformed into hiSCs by two TFs. WT1 and SOX9 are two very important marker genes, loss of WT1 results in defect of Sertoli function and germ cell loss. In this study, the expression of endogenous WT1 and SOX9 in two TFs induced hiSCs need to be examined by immunostaining and other approaches.

3) The lipid droplet was stained with BODIPY, it is very hard to tell the morphology of lipid droplet. Oil Red O staining is suggested to examine the morphology of lipid droplet in hiSCs and control Sertoli cells.

4) SF1 is a very important TF for steroidogenic cell development and many genes which involved in steroidogenesis are induced by SF1. The authors only examined the expression of 3β-HSD, the expression of other genes also need to be examined in hiSCs.

Reviewer #3:

Liang et al. report experiments in which they have transdifferentiated human fibroblast cells into Sertoli-like cells (hiSCs) that are morphologically, physiologically, and transcriptionally similar to adult human Sertoli cells. Moreover, they successfully reduced the number of transcription factors necessary for cell reprogramming from five to two. The authors have displayed an incredible amount of work, and their results have considerable implications for infertility treatment in humans. However, this is a rehash of the mouse work with no conceptional advancement from what was done by the Church lab.

- Were all TFs in separate plasmids or were the RNAs being expressed from a single promoter. There is significant variability in 5TF expression in the IF data. Actually how many of the transformed cells express multiple TFs.

- In Figure 1D – there is no clear separation or conversion of HPF to AMH:EGFP+ cells. The dH1 induced Sertoli cells still look very much fibroblasty. Do these cells express *SOX9*?

- Although the authors demonstrate a strong trend of similarity between hiSCs and aSCs, their analysis nonetheless clearly demonstrates that differences remain between these groups (e.g., there are numerous transcriptomic differences visible in Figure 2). However, the authors never detail what the differences between hiSCs and aSCs are.

- The PCA plot Hierarchical clustering needs to be shown when comparing 5TF, 2TF, vs. aSCs. The authors opted not to include the data.

- aSCs make for an important positive control, but a direct comparison between aSCs and hiSCs is often missing (e.g. co-plating isolated spermatogonia with SCs; human IL-2 assay (Figure 6C); luciferase assay in xenotransplants; and CX43 deletion experiments).

- Methods on germ cell/Sertoli cell coculture need to be further detailed – what strain of mice was used for SSC culture. A two day culture is not sufficient to show the 2F Sertoli cells can support germ cell maintenance – comparing this to aSC is important.

- In the CX43 experiment it’s not clear why the authors think loss of CX43 affect Sertoli cell specification when in humans the testis has Sertoli cells but actually males don't have germ cells. Also it's not clear why they would see an increase in reprogramming efficiency or decrease when overexpression. It is unclear how the authors concluded that CX43 deletion lead to dedifferentiation of Sertoli cells to a less mature state (subsection “CX43 deletion disrupts gap junctions and alters the expression profile of hiSCs”, second paragraph). If you mix with germ cells can the Sertoli cells differentiate. why not stain with SOX9 – this is not affected by maturation.

- It was frequently unclear how many cells were used in a given replicate or how many replicates were included in an experiment.

- Authors should use consistent labeling in their graphs (e.g. "2K7" vs. "CTRL") and use the same scale (and ideally range) for their axes. The scale of the y-axes in Figure 4B are confusing.

---

## [Author Response]

Reviewer #2:[…] 1) In this study, AMH was used as a reporter to identify the hiSCs. It has been reported that AMH is abundantly expressed in mouse Sertoli cells during embryonic stage and AMH protein is not detected in Sertoli cells after birth. The expression of AMH is probably different from other species. It is necessary to examine the expression of AMH in human Sertoli cells by immunostaining. The specificity of AMH promoter also need to be verified by transfecting human adult Sertoli cells.

AMH expression is expressed in adult human Sertoli cell although the expression may be relatively lower than embryonic Sertoli cells. This is supported by the recent single-cell sequencing study of adult human testis by an independent research group (Wang et al., 2018). They concluded that AMH is specifically expressed in Sertoli cells but not in other somatic cells including Leydig cells of adult human testis. (See Figure 1D from Wang et al., 2018).

We have also confirmed the specificity of AMH promoter and expression of EGFP in our reporter system by transfecting the vector into human adult Sertoli cells (Figure 1—figure supplement 3).

2) The authors claim that human fibroblasts could be transformed into hiSCs by two TFs. WT1 and SOX9 are two very important marker genes, loss of WT1 results in defect of Sertoli function and germ cell loss. In this study, the expression of endogenous WT1 and SOX9 in two TFs induced hiSCs need to be examined by immunostaining and other approaches.

We examined the transcriptional level of WT1 and SOX9 by RNA-seq and found that WT1 expression in both aSCs and 2F-hiSCs was relatively low but SOX9 expression in both aSCs and 2F-hiSCs was similar (Figure 3—figure supplement 1). When we tried to immunostain WT1, we were not able to detect the proteins in the nuclei of both cell types (data not shown). However, we detected similar level of SOX9 in the nuclei of aSCs and 2F-hiSCs (Figure 3—figure supplement 3). The immunostaining results are consistent with the RNA-seq results. We speculate that WT1 may be important during in vivo development of Sertoli cells, but it may not be important during in vitro reprogramming of fibroblasts to Sertoli cells.

3) The lipid droplet was stained with BODIPY, it is very hard to tell the morphology of lipid droplet. Oil Red O staining is suggested to examine the morphology of lipid droplet in hiSCs and control Sertoli cells.

We have used Oil Red O staining to examine the existence of lipid droplet and found that red lipid droplets appeared in both aSCs and 2F-hiSCs but not in control cells (Figure 4—figure supplement 1).

4) SF1 is a very important TF for steroidogenic cell development and many genes which involved in steroidogenesis are induced by SF1. The authors only examined the expression of 3β-HSD, the expression of other genes also need to be examined in hiSCs.

Beside 3β-HSD, we have examined the transcriptional level of many other genes expressed in steroidogenic Leydig cells. We found that most of these genes (8 out of 10) are not detectable or expressed at very low level (Figure 3—figure supplement 4B). Two of these genes were detectable (STAR and PGD), but they are also expressed in aSCs.

Reviewer #3:[…] The authors have displayed an incredible amount of work, and their results have considerable implications for infertility treatment in humans. However, this is a rehash of the mouse work with no conceptional advancement from what was done by the Church lab.- Were all TFs in separate plasmids or were the RNAs being expressed from a single promoter. There is significant variability in 5TF expression in the IF data. Actually how many of the transformed cells express multiple TFs.

Each TF was cloned separately into lentiviral plasmid so each gene was expressed from its own promoter (human EF1alpha). Figure 1A showed the fibroblasts transduced with lentivirus of each TF plasmid. The first experiment aim was to confirm that the transduced fibroblasts expressed high level of each TF in the expected subcellular location, nucleus. When we mixed all five TFs to obtain AMH:EGFP+ cells, we detected high expression of all five TFs (Figure 3—figure supplement 1).

- In Figure 1D – there is no clear separation or conversion of HPF to AMH:EGFP+ cells. The dH1 induced Sertoli cells still look very much fibroblasty. Do these cells express SOX9?

After TF induction, AMH:EGFP+ cells from both HPF and dH1 fibroblasts undergo morphological changes. This was shown in Figure 1E. In contrast to dH1 transduced with p2k7 control cell that showed fibroblast morphology, AMH:EGFP+ cells became more epithelial-like morphology. These cells expressed SOX9 as shown in Figure 3—figure supplement 1 (RNA expression) and Figure 3—figure supplement 3 (protein expression).

- Although the authors demonstrate a strong trend of similarity between hiSCs and aSCs, their analysis nonetheless clearly demonstrates that differences remain between these groups (e.g., there are numerous transcriptomic differences visible in Figure 2). However, the authors never detail what the differences between hiSCs and aSCs are.

We compared the characteristics of hiSCs to aSCs and used aSCs as a positive control. Indeed, more of the characteristics are similar between these two cell types. There were some minor differences such as expression level of some genes as shown in Figure 3—figure supplement 1. However, the overall transcriptional profiles are similar as in Figure 2D and PCA analysis in Figure 3—figure supplement 2.

- The PCA plot Hierarchical clustering needs to be shown when comparing 5TF, 2TF, vs. aSCs. The authors opted not to include the data.

We have added PCA plot in Figure 3—figure supplement 2 and showed that 2F-, 5F-hiSCs and aSCs clustered closely but distanced away from fibroblasts.

- aSCs make for an important positive control, but a direct comparison between aSCs and hiSCs is often missing (e.g. co-plating isolated spermatogonia with SCs; human IL-2 assay (Figure 6C); luciferase assay in xenotransplants; and CX43 deletion experiments).

In the previous version of manuscript, we used aSCs in some experiments including xenotransplants (Figure 6—figure supplement 1B). In the revised manuscript, we have added more control experiments using aSCs, including co-plating with isolated spermatogonia (Figure 5—figure supplement 2), IL-2 assays (Figure 6—figure supplement 1C). We are not able to create CX43 deletion cell lines in aSCs because they are primary cells and not proliferative. In addition, the original goal of CX43 experiment is to determine whether CX43 is important during Sertoli cell formation or induction. aSCs is not an ideal positive control in this experiment.

- Methods on germ cell/Sertoli cell coculture need to be further detailed – what strain of mice was used for SSC culture. A two day culture is not sufficient to show the 2F Sertoli cells can support germ cell maintenance – comparing this to aSC is important.

We described the information about the mouse strain (C57BL/6) we used in the SSC experiments in the Materials and methods section under ‘Co-transplantation of hiSCs with 293FT cells’.

- In the CX43 experiment it’s not clear why the authors think loss of CX43 affect Sertoli cell specification when in humans the testis has Sertoli cells but actually males don't have germ cells. Also it's not clear why they would see an increase in reprogramming efficiency or decrease when overexpression. It is unclear how the authors concluded that CX43 deletion lead to dedifferentiation of Sertoli cells to a less mature state (subsection “CX43 deletion disrupts gap junctions and alters the expression profile of hiSCs”, second paragraph). If you mix with germ cells can the Sertoli cells differentiate. why not stain with SOX9 – this is not affected by maturation.

First, we observed the expression of several markers of immature Sertoli cells was higher and the expression of markers of mature Sertoli cells was lower in CX43KO than in the WT hiSCs (Figure 7I). Second, a previous study reported an association between Sertoli-only syndrome (SCO) and the lower mRNA expression of CX43 (Defamie et al., 2003) and suggested that the absence of CX43 rendered the Sertoli cells more immature.

AMH is a gene expresses higher in immature Sertoli cells and we used AMH:EGFP+ to measure the reprogramming efficiency. Hence, the fact that CX43KO cell lines yielded higher AMH:EGFP+ percentage is consistent with the RNA-seq result showing that CX43KO cells expressed more immature Sertoli markers.

- It was frequently unclear how many cells were used in a given replicate or how many replicates were included in an experiment.

We have indicated the number of cells and number of replicates in the figure legends. Please let us know which number is missing and we will add the information.

- Authors should use consistent labeling in their graphs (e.g. "2K7" vs. "CTRL") and use the same scale (and ideally range) for their axes. The scale of the y-axes in Figure 4B are confusing.

We apologized that the labeling of control were different in different figures. We were considering to change all labels as CTRL, but there are more than one control cell type in certain sets of experiments, e.g. Figure 3D. Hence, we went through the figures and added more information to the controls whenever it is not clear. Please let us know if there is any control we need to clarify.